# Charge-specific size-dependent separation of water-soluble organic molecules by fluorinated nanoporous networks

Jeehye Byun[1], Hasmukh A. Patel[1,†], Damien Thirion[1] & Cafer T. Yavuz[1,2]

Molecular architecture in nanoscale spaces can lead to selective chemical interactions and separation of species with similar sizes and functionality. Substrate specific sorbent chemistry is well known through highly crystalline ordered structures such as zeolites, metal organic frameworks and widely available nanoporous carbons. Size and charge-dependent separation of aqueous molecular contaminants, on the contrary, have not been adequately developed. Here we report a charge-specific size-dependent separation of water-soluble molecules through an ultra-microporous polymeric network that features fluorines as the predominant surface functional groups. Treatment of similarly sized organic molecules with and without charges shows that fluorine interacts with charges favourably. Control experiments using similarly constructed frameworks with or without fluorines verify the fluorine-cation interactions. Lack of a $\sigma$-hole for fluorine atoms is suggested to be responsible for this distinct property, and future applications of this discovery, such as desalination and mixed matrix membranes, may be expected to follow.

[1] Graduate School of Energy, Environment, Water and Sustainability, Korea Advanced Institute of Science and Technology, Daejeon 305-701, Republic of Korea. [2] Department of Chemistry, Korea Advanced Institute of Science and Technology, Daejeon 305-701, Republic of Korea. † Present address: Department of Chemistry, Northwestern University, Evanston, Illinois 60208, USA. Correspondence and requests for materials should be addressed to C.T.Y. (email: yavuz@kaist.ac.kr).

Materials with nanopores (pore diameters under 100 nm) present unprecedented opportunities, owing to their modular synthetic routes and potential in heterogeneous catalysis[1,2], gas separation[3,4], water treatment[5–7], charge carrier[8] and sensors[9]. The size and shape of the pores found within the networks, combined with target-specific functionality, facilitate selective separation of the desired sorbate molecules[10], for example, hydrocarbon and gas separations using metal organic frameworks[11–15], porous organic cages[16], zeolites[17,18], covalent organic polymers (COPs)[19], polymers of intrinsic microporosity[20,21], activated carbons[22,23] and several other nanoporous materials[24–28].

Water resources are under threat from soluble, man-made organic pollutants such as pesticides, artificial dyes and medicines, particularly because of the increased demand for industrial water consumption and waste generation[29]. Most water filters perform adequately with insoluble or slightly soluble, large organic species, but the removal of highly water-soluble organic pollutants with fine dimensions (especially sub nm) is very challenging[30,31]. Common sieving methodologies that are used by the gas separation media[32] may not always work since the presence of strong intermolecular forces in aqueous media requires new approaches such as charge-specific size-dependent separations. Despite a few attempts for the removal of organic molecules using nanoporous adsorbents[33–35], size-dependent separation and the related chemistry of the interaction between these molecules and sorbents had not been well examined. For example, Li et al. reported size-dependent separation of organic molecules using metal organic frameworks[36], however, the experiments were carried out in a non-aqueous solvent (dimethyl formamide), raising questions of practicality for water treatment. In addition, all substrates were inherently charged. More recently, Alsbaiee et al. published a study on cyclodextrin based porous polymers (P-CDP) for contaminant removal[37]. They have not observed a size-dependent or a charge-specific interaction, of which we suspect the pore size distributions (1.8–3.5 nm) being large enough to accommodate all tested substrates, similar to the activated carbon that is tested in this study. Yang et al. also reported superhydrophobic fluorine containing, microporous (<2 nm) polymers with high uptake capacities for organic dyes and lead[38]. Perhaps, lack of appropriate sorbents that feature well-defined pore geometry, sorbate selective functionality and most importantly, stability in water is the limiting factor for efficient size-dependent separation of organic contaminants from polluted water.

Porous network polymers, such as COPs[39], show great potential for investigating the size-dependent separation of the water-soluble organic molecules since they are robust and can be readily functionalized by pre- or post-synthetic methods[40]. These nanoporous COP structures with excellent water stability can be designed into targeted porosity and chemical functionalities according to the requirements of the sorbate species[41]. In this study, we have demonstrated a charge-specific size-dependent uptake of soluble organic molecules from water by micropore confinement using a tailored structure, COP-99, and the existing nanoporous structures, covalent triazine framework (CTF)[42], perfluorinated CTF[43] and activated charcoal norit (ACN), as controls. In addition to the sieving properties of microporous networks, we discovered that fluorine functionalities on the pore walls contribute to the selective uptake of charged species over similarly sized uncharged molecules. COP-99 is easily synthesized at scalable quantities through a catalyst-free, self-condensation of an inexpensive monomer. Our findings conclude that a combination of pore confinement and fluorine functionality is key for size and charge selective adsorption of soluble organic molecules, especially from water.

## Results

**Selection of a suitable functional group.** The main challenge for sorbent-based water treatment is the soluble organic species[44], precisely because the less soluble are easily removed by sedimentation or adsorption on hydrophobic surfaces such as activated carbons[45]. In search of a particular functionality that (1) is widely available and easily introduced, (2) would interact favourably with soluble organic species and (3) do not contain acidic protons, led us to fluorinated molecules. Fluorine is not only the strongest of the electronegative atoms, it also does not do a 'halogen bonding' (Supplementary Fig. 1), in which halogens (except fluorine) form a $\sigma$-hole and interact with electron donating groups in a similar fashion to substituted hydrogens at a hydrogen bonding[46]. These features, plus the well-known hydrophobicity make fluorine an attractive functional group for adsorptive treatment of soluble organic species. To the best of our knowledge, fluorine was never considered in such critical applications before this study.

**Synthesis of a fluorinated covalent organic polymer.** For a sustainable synthesis of highly fluorinated adsorbent, we used an inexpensive, commercially available monomer (can be purchased from over 55 providers), tetrafluorohydroquinone (TFHQ) to make the network polymer (indexed as COP-99) through a unique, base-promoted, and catalyst-free self-condensation reaction. The synthesis follows an effective nucleophilic substitution based network polymer formation (Fig. 1a)[47]. The COP-99 is generated by the $S_N2$ reaction between fluorine and deprotonated hydroxyl terminal functionality, and owing to stoichiometric inferiority of –OH units on TFHQ (2:4 = OH:F), the final product has an excess of dangling fluorines. The measured fluorine content in COP-99 was 21.5 wt% (0.52 equivalent to the fluorine on TFHQ), which is comparable with that of highly fluorinated porous polymers (Supplementary Table 1). Due to the highly cross-linked network structure, COP-99 was insoluble in common solvents (Supplementary Fig. 2a) and boiling water (Supplementary Fig. 2d–f and Supplementary note 1).

COP-99 exhibits amorphous nature as reflected by its powder X-ray diffraction pattern, and the morphology of COP-99 from scanning electron microscopy observation was highly granular with mostly irregular shapes (Fig. 1b and inset). Thermogravimetric analysis of COP-99 revealed high thermal stability of up to 300 °C without any noticeable mass change (Supplementary Fig. 3). The XPS spectra, $^{19}$F solid-state nuclear magnetic resonance (NMR) spectrum and fourier transform infrared spectroscopy (FT-IR) spectrum demonstrated that the structural composition of COP-99 was originated from the expected random polymerization. The C 1s spectrum of COP-99 (Fig. 1c) can be broken into three peaks with binding energies at 284.6 eV, 286.6 eV and 286.9 eV, which can be designated to the aromatic sp$^2$ carbon, ether C–O, and covalent C–F carbon, respectively. The O 1s spectrum (Fig. 1d) contains three peaks at 531.5 eV, 532.8 eV and 534.1 eV, attributed to unreacted –OH, ether C–O–C and C–O bond, respectively. FT-IR spectrum of COP-99 also displays the presence of hydroxyl units and ether linkages with intense band at 3,670 cm$^{-1}$ and 1,030 cm$^{-1}$ (Supplementary Fig. 4). The single peak at 687.6 eV in the F 1s spectrum corresponds to the aromatic C–F bond (Fig. 1e), and the FT-IR vibration at 740 cm$^{-1}$ is also matching well to the existence of the C–F bond (Supplementary Fig. 4). Figure 1f shows that the $^{19}$F NMR spectrum of COP-99 exhibits the signals at about $-100.6$ p.p.m., $-109.59$ p.p.m. and $-125.06$ p.p.m., indicating that the fluorines are located on benzene ring in three different asymmetric positions.

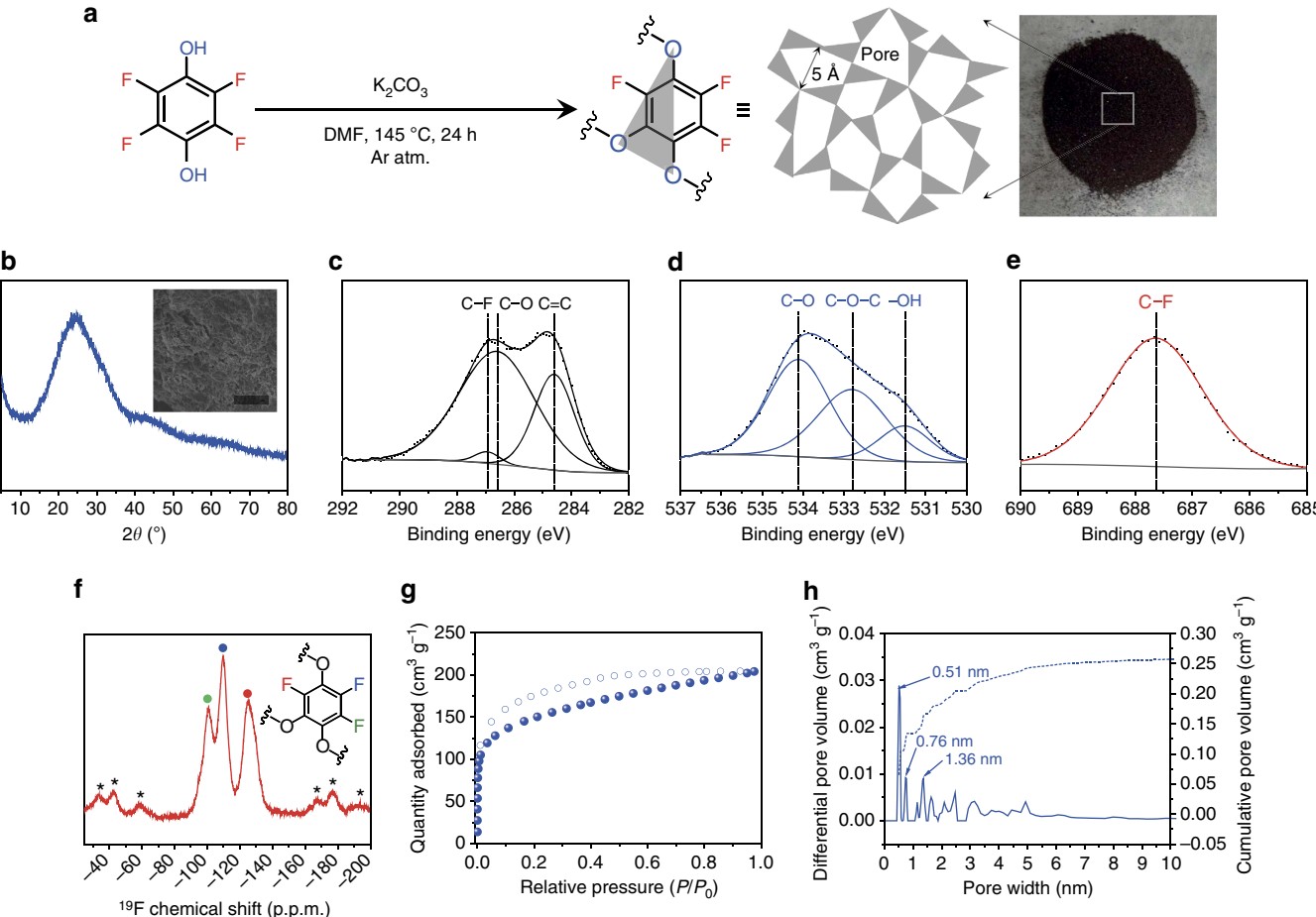

**Figure 1 | Synthesis and characterization of COP-99.** (**a**) Proposed chemical structure of COP-99 along with its synthetic procedure. The obtained polymer was dark brown powder. (**b**) Powder X-ray diffraction pattern of COP-99. Inset displays scanning electron microscopy image of COP-99. Scale bar, 10 μm. (**c**) C 1s, (**d**) O 1s, and (**e**) F 1s XPS spectra of COP-99. (**f**) Solid-state $^{19}$F-NMR spectrum of COP-99. Black asterisks (*) indicate spinning side bands. (**g**) Argon adsorption–desorption isotherm of COP-99 measured at 87 K and (**h**) corresponding NLDFT pore size distribution.

Despite the randomness in the formation of COP-99, each synthetic batch yields an ultra-microporous structure with 3.0% surface area calculation distribution from three separate measurements. To put into perspective, a size distribution value <5% is commonly considered highly monodisperse in nano materials[48]. Surface area analyses were done by Ar physisorption measurements at 87 K (Fig. 1g). We chose Argon over $N_2$ isotherms to negate any potential interaction of the probe gas with the aromatic units of COP-99, since we previously observed that certain functionalities, such as azo units could show unexpected behaviour against nitrogen molecules[19]. COP-99 is ultra-microporous and exhibits a hysteretic *type I* sorption isotherm, pointing to sieving characteristics. The Brunauer–Emmett–Teller (BET) specific surface area of COP-99 was calculated to be $479 \, m^2 \, g^{-1}$ and total pore volume was $0.262 \, cm^3 \, g^{-1}$ (Supplementary Fig. 5). These values put COP-99 in league with zeolites, which have typically surface areas of $300–700 \, m^2 \, g^{-1}$. The microporosity ($V_{micro}/V_{total}$, where $V_{micro}$ is micropore volume and $V_{total}$ is total pore volume) of COP-99 was 0.7, implying that the majority of pores are within micropore region. The non-local density functional theory (NLDFT) pore size distribution reveals that COP-99 has a predominant pore size of 0.51 nm with two secondary bimodal pores at 0.76 and 1.36 nm. (Fig. 1h). As expected, the microporous COP-99 with high fluorine content showed good $CO_2$ sorption capability (Supplementary Fig. 6 and

Supplementary note 2), and this further proved that COP-99 has sufficient porosity for the adsorption of small molecules. The simple, one-pot, catalyst free, self-condensation of TFHQ under ambient conditions, therefore, successfully produces the ether-bridged fluorinated COP-99, and such a high fluorine content and surface area of the COP-99 would be appropriate for a size-dependent adsorption study.

**Charge-specific size selective adsorption in aqueous phase.** To demonstrate a selective substrate separation from water, we treated COP-99 with dye molecules (Fig. 2), specifically methylene blue (MB), rhodamine B (RDB) and brilliant blue G (BBG). The substrates are selected by their representation of different size regimes, easy spectroscopic analysis and exceptional solubility in water. As shown in Fig. 2a–c, three dyes are different in their size, and the calculated van der Waals diameters are given as a range on a scale where min and max subscripts indicate minimum and maximum projection diameter of dyes, respectively (see Supplementary Fig. 7 for the detailed diameter values). The maximum size difference among the three dyes is found to be 1.4 nm, calculated from $BBG_{max}$ to $MB_{min}$. Dye adsorption was conducted via monitoring the change of dye concentration in regular time interval by ultraviolet–visible spectrophotometer.

As it is expected, plenty of fluorines in COP-99 make the structure hydrophobic. Water contact angle measurement shows

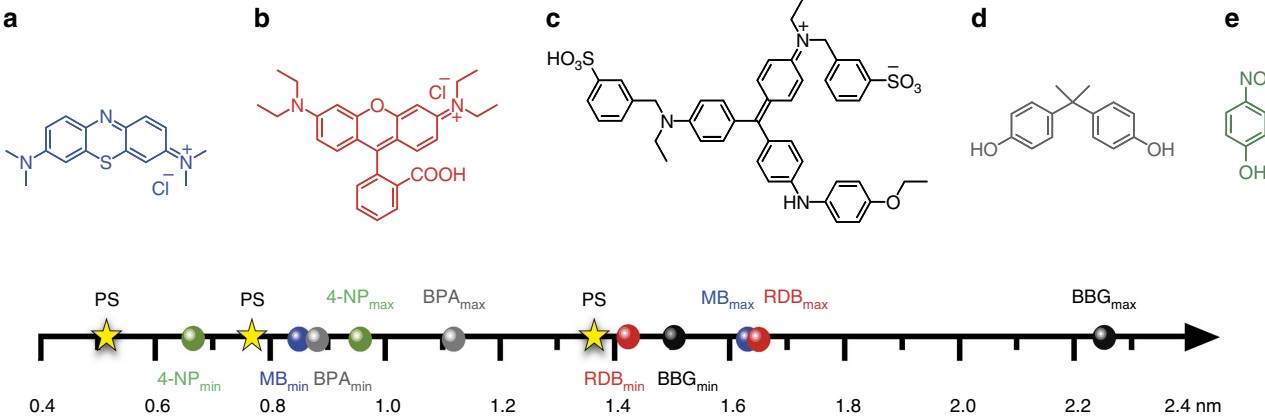

**Figure 2 | Substrates that are tested in charge-specific size-dependent separation study.** Chemical structure of dye molecules and their calculated van der Waals diameters in a scale bar; (**a**) MB (blue sphere on scale bar), (**b**) RDB (red sphere), (**c**) BBG (black sphere), (**d**) BPA (grey sphere) and (**e**) 4-NP (green sphere). The min and max subscripts indicate minimum and maximum projection diameter of dyes, respectively. pore size (star) indicates three accessible pore sizes existing in COP-99.

that COP-99 exhibits surface hydrophobicity with a water contact angle of 142° (Supplementary Fig. 2b). If added to water, COP-99 floats on the surface (Supplementary Fig. 2c). Hydrophobic nature of polymeric adsorbent is also known to be helpful for adsorption of organic molecules[38]. A simple dispersion of COP-99 into a MB solution without agitation clearly shows a colour change from pale blue to transparent, indicating the capability of COP-99 for separating organic molecules (Fig. 3a). The COP-99 was then subjected to a systematic dye adsorption test with the three dye molecules with varying sizes (Fig. 2a–c). Through spectroscopic monitoring of MB, RDB and BBG with COP-99, we found size-dependency when the size of the substrate was systematically increased. MB was completely removed in 3 h of interaction with COP-99 (Fig. 3b), while the larger RDB and BBG did not exhibit any noticeable change in their concentration for the same time period (Supplementary Fig. 8). Such a size-dependent separation of dye molecules could be resulted from molecular sieving effect of the microporous network[35]. As displayed in Fig. 2, the size of RDB and BBG is apparently larger than the accessible pore size of COP-99, and the MB_min is only fitting within the pore size of COP-99, leading to size-exclusive behaviour.

It is conceivable that the origins of the size selective adsorption capability of COP-99 is primarily dependent on sieving properties, where pore openings limit the size of the substrate that penetrates deeper into the network. This, however, does not explain the steps in the MB absorption isotherm (Fig. 4a). In a pore limited diffusion scenario, kinetics is equally hindered (hence linear curve with a constant slope) for all the substrate species provided that the wall surfaces do not interact favourably with the incoming guests. The favourable interaction between adsorbate and adsorbent should, therefore, affect the adsorption kinetics. The two concepts are best showcased in Long *et al.*[49], where hexane isomers were shuttled inside a metal organic framework in a stepwise fashion, creating pseudo saturations, akin to monolayer—multilayer coverage switching in hierarchical porous systems. Since there are also noticeable plateaus, we suspected that there might be favourable interactions of fluorines with MB. The specific interaction between fluorine on COP-99 and MB molecule was confirmed by FT-IR spectrum (Supplementary Fig. 4b). After the MB adsorption, COP-99 showed insignificant change in FT-IR spectrum due to the weak electrostatic interaction between COP-99 and MB molecule.

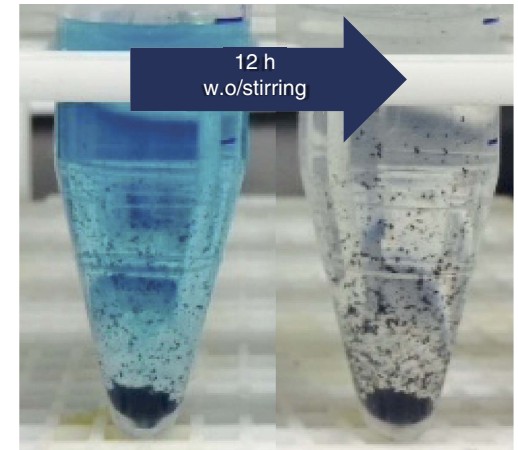

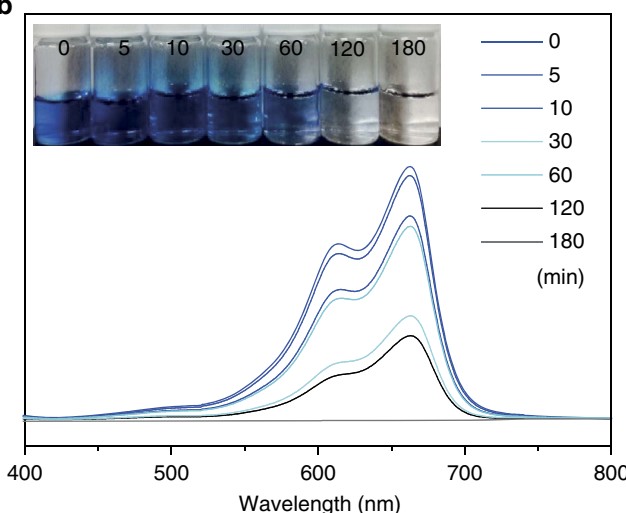

**Figure 3 | Monitoring the organic molecule uptake of the nanoporous network.** (**a**) Colour change of MB solution by soaking with COP-99 overnight. (**b**) ultraviolet–visible absorption spectra of aqueous MB treated with COP-99 at different interval. The inset photograph shows the corresponding colour change of the MB solutions.

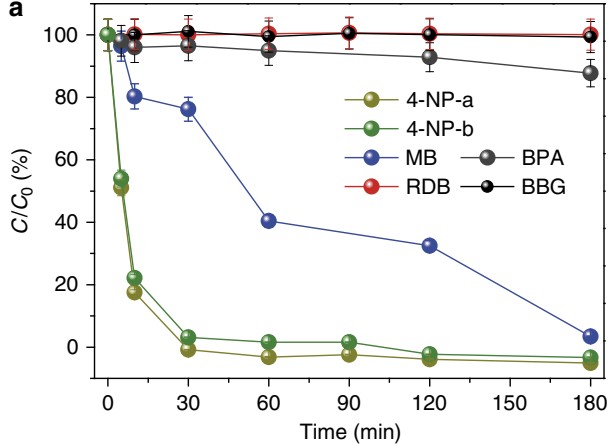

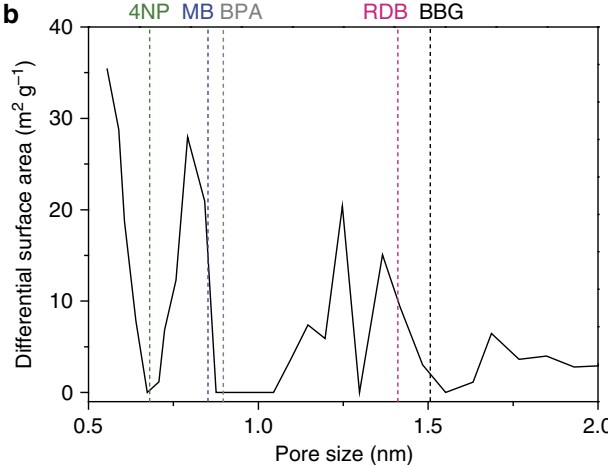

**Figure 4 | Size selective separation of water-soluble substrates by COP-99.** (**a**) Change in dye concentrations over time after being treated with COP-99 in terms of absorbance relative to initial absorbance ($C/C_0$). Initial concentration ($C_0$) of all the dyes was adjusted to be 50 μM. 4-NP was tested both in acidic (4-NP-a, pH = 4) and basic (4-NP-b, pH = 9) conditions. Error bars reflect s.d. from duplicates. (**b**) Surface area distribution with respect to pore size, and its relation towards minimum van der Waals size of dye molecules.

Nonetheless, there was one minor peak disappearing through the MB adsorption, which is at the position for F-phenyl ring bending mode at $740\,cm^{-1}$. This may be attributed to the hindrance of the C–F bond angle as the MB molecule favourably binds on the fluorine of COP-99.

**Fluorine-cation interaction for charged molecules separation.** Since all three dye molecules we tested were inherently charged, we decided to study an uncharged, MB-size bisphenol A (BPA) (Fig. 2d) as a control to understand whether there is any unusual fluorine-cation interaction prevalent in the current adsorption scenarios. Under the identical batch conditions with MB, BPA was hardly adsorbed by COP-99, which shows as low as 12% of separation efficiency even after 3 h of agitated interaction (Fig. 4a). This was unexpected since around these sizes the movement of the guest molecules would be hindered (slow diffusion) but not completely altered. Our results, therefore, show the first time that a charged cation on a guest molecule assists considerably in the selective uptake when molecular size approached the pore-opening widths of the fluorinated porous host, in this case, the COP-99.

Knowing that studying in aqueous mediums has the benefit of shuffling charged species of protic molecules by simply varying pH, we have studied a smaller but amphoteric molecule with a dual nature, to understand how dominant the role of charges would be on a selective uptake. The 4-nitrophenol (4-NP) is a smaller dye than MB (Fig. 2e), and has a p$K_a$ of 7.2. It can be exclusively neutral at acidic pH but dissociates to anionic form by increasing pH to basic conditions[50,51]. This test would especially show us that which factor (size versus charge) is predominant in the size selective adsorption.

An aqueous solution of 4-NP was therefore prepared to be acidic (pH 4) and basic (pH 9) for switching between charged and uncharged states. The test solutions are designated as 4-NP-a (acidic) and 4-NP-b (basic). Upon testing COP-99 with 4-NP-a and 4-NP-b, we observed almost no hindrance in uptake; COP-99 shows 99 and 97% of removal efficiency in 30 min for 4-NP-a and 4-NP-b, respectively (Fig. 4a). This result points out to the contributions by size considerations more so than the fluorine-charge interactions, as the size of the target molecules is small enough to penetrate the pore network. It is rather expected that the guest molecules need not fully interact with the pore openings and walls, if their sizes are much smaller. The smaller molecule, however, was always taken faster owing to the size-dependent adsorption capability of COP-99. For instance, NP isomers (2-NP and 3-NP), having a slight size difference with 4-NP, also showed a complete removal in the given time. Among three, the 2-NP with smallest size, exhibited the fastest adsorption kinetics (Supplementary Fig. 9). The small-sized 4-NP molecule also exhibited reversible adsorption–desorption behaviour on COP-99. Throughout a repeated cycle, COP-99 showed excellent uptake of 4-NP-a with negligible loss of removal efficiency after 6th cycle (Supplementary Fig. 10).

One might consider that the selective uptake of the adsorbates could originate from hydrogen bonding between the acidic protons of the guest molecules and the fluorines of COP-99. To study the hydrogen bonding effect in the adsorption, we tested m-phenylenediamine (m-PD) with two amino groups. The m-PD has a theoretical maximum size of 0.885 nm, close to the 2-NP, so that the adsorption of m-PD was compared with that of 2-NP. Because of its small size (compared to the pore openings), the m-PD was completely removed after 1 h of adsorption in both acidic and basic conditions (Supplementary Fig. 11). Owing to the protonation of amino group, m-PD-a showed a bit faster adsorption behaviour than the m-PD-b counterpart, exhibiting 72 and 60% at the first 10 min of interaction with COP-99, respectively. Compared to the 2-NP-a, however, the adsorption kinetics of m-PD was slower, implying that the hydrogen-bonding effect was not as dominant as expected since there is more hydrogen bonding contribution in m-PD. This is in line with the low uptake of BPA using COP-99, where the possible hydrogen bonding on two hydroxyl units does not dramatically affect the adsorption efficiency (Fig. 4a).

This size selectivity is in excellent agreement with the pore size distribution based separations[52]. As displayed in Fig. 4b, three structures within the pore size distribution range of COP-99, 4-NP, MB and BPA, have shown some degree of adsorption via interacting with COP-99 depending on their size and charge. On the other hand, RDB and BBG, which are larger than the average pore openings–despite the favourable fluorine-charge interaction–, did not show any uptake for 3 h of interaction. The latter observation further confirms that synergetic effect between narrow pore confinement and fluorine-cation interaction leads to a charge-specific size-selective adsorption behaviour. To further elucidate these points, we iterated that when the size of sorbate molecule exceeds the pore size threshold of COP-99, the adsorption does not go beyond surface coverage in spite of the

inherent charge of the molecules. We, therefore, conducted long-term adsorption test using RDB and BBG, which showed no uptake for 3 h of interaction, to confirm whether those molecules can occupy the pore voids over a long-term adsorption on COP-99 (Supplementary Fig. 12). After a 48 h of adsorption, RDB and BBG exhibited 27 and 3% of uptake on COP-99. Since the theoretical minimum size of RDB and BBG is slightly bigger than the pore threshold of COP-99, there is a chance of being taken in COP-99 for a long-time interaction. In particular, RDB with a size much closer to the average pore size of COP-99 could be adsorbed, showing a low but noticeable uptake of 27%. These size-dependent observations may be attributed to the surface adsorption where RDB is packed in more finely than BBG.

**Proof for fluorine-cation interaction in control structures**. Size and charge selective separation behaviour of COP-99 prompted us to look for similar porous structures to verify our observations. We realized that two previously reported porous polymers, CTF (ref. 42) and fluorinated CTF (F-CTF)[43] have similar pore size distributions with COP-99, and in F-CTF, feature fluorines. The CTF and F-CTF are therefore adopted as platforms to confirm the effect of surface fluorination for separating charged molecules.

We synthesized the CTF and F-CTF based on the ionothermal reactions of corresponding nitriles[42]. Porosity of CTF and F-CTF were analysed by Ar sorption isotherms at 87 K (Fig. 5a) and agree well with the literature. These nanoporous polymers exhibit *type I* isotherms, typical for microporous materials. The BET specific surface areas for CTF and F-CTF were found to be $863.5\,m^2\,g^{-1}$ and $1{,}342.9\,m^2\,g^{-1}$, respectively, and the pore volume analysed at $P/P_0 = 0.975$ was $0.38\,cm^3\,g^{-1}$ and $0.59\,cm^3\,g^{-1}$, respectively (Supplementary Fig. 5). Pore size distribution of CTF and F-CTF was identical, mainly due to the

structural similarity of CTF and F-CTF. The NLDFT pore size distribution displayed bimodal average pore size of 0.51 and 1.04 nm for both nanoporous polymers (Fig. 5b).

Under a constant molar concentration of dyes (50 μM), CTF and F-CTF show distinct size-dependent adsorption behaviour–similar to COP-99, and the degree of separation has greatly reduced as the size of charged dye molecules increases (Fig. 5c,d). In the case of the small sized and charged MB, CTF exhibited ∼50% of removal efficiency in 360 min, whereas F-CTF showed complete separation after only 40 min of interaction. Even for RDB and BBG, dyes with larger van der Waals sizes, CTF displayed almost no removal efficiency while F-CTF show slight uptake of RDB and BBG with separation efficiency of 24 and 14%, respectively, under identical conditions. When the size of sorbate molecules is small enough, that is, 4-NP-a and 4-NP-b, both CTF and F-CTF readily capture the dyes (Supplementary Fig. 13). CTF and F-CTF exhibited much faster uptake of 4-NP-a and 4-NP-b than COP-99, mainly due to the larger pore size and higher surface area. Both CTF and F-CTF removed 99% of 4-NP-a within 4 and 1 min, while CTF and F-CTF captured about 99 and 95% of 4-NP-b for 12 min of adsorption, respectively. The uptake is also noticeably faster in F-CTF, mainly due to the interaction of fluorines with charged molecules.

In support of our discovery of size and charge-dependent properties of fluorinated networks, it is very clear that the F-CTF, with almost identical pore size distribution as CTF (Fig. 5b), shows much higher and faster uptake for MB. This difference is exclusively because of the fluorinated surface of F-CTF interacting readily with the cationic ammonium centre on MB. As observed in COP-99, facile interaction between fluorine and charged centres of the soluble dye molecules are responsible for this enhanced adsorption. F-CTF, therefore, enhances the interaction of the surface with the cationic site in MB, in a way that a concerted railing mechanism may be suspected.

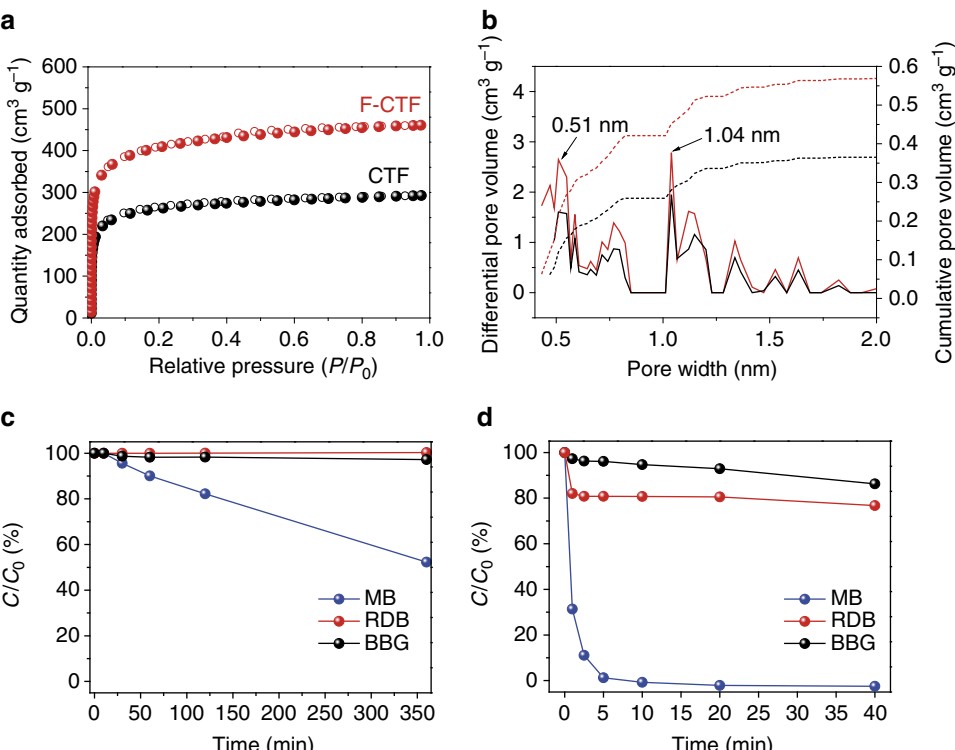

**Figure 5 | Nanoporous CTFs as control structures with or without fluorines.** (**a**) Argon adsorption–desorption isotherms of CTF and F-CTF measured at 87 K, and (**b**) corresponding NLDFT pore size distribution. Change in dye concentrations over time after being treated by (**c**) CTF and (**d**) F-CTF. Initial concentration ($C_0$) of the dyes was adjusted to be 50 μM.

**Zeta potential measurements**. Surface charges of sorbents are known to play a significant role while separating charged contaminants; therefore, we studied the zeta potential characteristics of these nanoporous polymers. The zeta potential of COP-99, as expected, exhibits negative due to the existence of electronegative fluorines (Fig. 6a). The COP-99 dispersion was stable throughout pH 5–9 with the value of c.a. −40 mV, and became least stable under highly acidic (pH 3.2) or basic conditions (pH 10.8). Owing to unreacted hydroxyl units in COP-99 that are shown in XPS analysis (Fig. 1d), the surface charge was affected by the protonation/deprotonation of the –OH groups in highly acidic and basic conditions.

CTF and F-CTF revealed distinct difference in surface charge properties, mainly because of the presence of fluorinated functionality in F-CTF (Fig. 6b). The zeta potential of CTF was positive at pH 3–9, varying from 11 to 38 mV, with the isoelectric point at around pH 11. However, the zeta potential of F-CTF was negative throughout the experimental pH conditions with the maximum value at −35.7 mV at pH 11. The zeta potential of F-CTF clearly indicates effect of the fluorine functionality which changes the surface negative over the pH range of 3–11, owing to strongly polar C–F bond[53].

Negative zeta potentials of both COP-99 and F-CTF confirm the fluorine-charge interaction, although the size limiting pore confinement (and sieving) based separation leads to slightly lower uptake in COP-99. We, therefore, reaffirm that fluorines with fine-tuned pore openings are key to selectively capture soluble organic species from water.

**Comparison with activated carbons**. To further confirm charge-specific size-dependent selectivity, we turned to ACN, an industrial standard adsorbent. As depicted in Supplementary Fig. 14a and b, ACN shows 895.5 m$^2$ g$^{-1}$ of specific BET surface area with distinct pore size of 0.5, 0.8 and 1.39 nm. Despite the microporosity, the pore size distribution is noticeably broad enough for all tested substrates. ACN also exhibits negative surface zeta potential values from pH 3.6 to pH 9, with the maximum value of −33.8 mV at pH 8.3 (Supplementary Fig. 14c), owing to the presence of functional groups such as hydroxyl, carboxyl and carbonyl group upon activation[54]. With the wide enough pores, the ACN adsorbs small-sized molecules, that is, 4-NP, MB and BPA, and all three dye solutions became clean within 10 min of interaction. The larger molecules were also removed by ACN in which the RDB was fully adsorbed in 2 h and BBG showed above 50% of removal efficiency in the first 3 h. The less size selectivity may be originated from the larger pore size with wider pore size distribution. These observations conclude that the pore opening has to be in the right size range for an effective separation and demonstrate that the narrow micropores and the fluorine functionalities are very effective when combined for size and charge selective adsorption of small organic molecules.

**Conceptual feasibility test**. Charge-specific size-dependent adsorption capability of COP-99 made us to test its feasibility under the conditions close to the actual pollution sources. Since the organic water contaminants are in low concentrations as a mixture of molecules with different sizes and charges[55], we conducted two conceptual feasibility tests for COP-99: (1) low concentration uptake of the adsorbates and (2) column separation of two mixed molecules.

First, we carried out an adsorption test of 4-NP in p.p.b. level and traced the change in concentration using high-performance liquid chromatography (HPLC), which enables us to detect 4-NPs in a more quantitative manner at these concentrations. The initial concentration of 4-NP was adjusted to be ∼100 p.p.b., and the adsorption was carried out under a mild acidic condition for 1 h (4-NP-a). The liquid chromatography (LC) spectrum of initial 4-NP-a solution showed a single sharp peak at 5.77 min of retention time, and after being treated with COP-99 for 1 h, the peak significantly diminished as shown at 5.75 min of retention time (Fig. 7a). From the analysis with the LC calibration curve, the actual concentrations of the initial and the treated solution were found to be 130.6 and 3.1 p.p.b., respectively, indicating the removal efficiency was ∼98%. This result agrees with the removal efficiency of 4-NP-a tested under higher concentration with ultraviolet–visible spectrum analysis. Other small molecules prepared in p.p.b. levels, that is, 4-NP-b, MB and BPA, also exhibited the similar patterns of adsorption as observed for adsorption in the high concentrations (Supplementary Fig. 15). Particularly, MB showed no residual peak on LC spectrum, indicating a complete removal after treatment with COP-99 for 3 h (Supplementary Fig. 15b).

The COP-99 was further utilized in a column experiment to separate two molecules having different sizes. We chose 4-NP and BBG mixture as their maximum absorbance wavelength in UV spectrum does not overlap each other. A column for the separation test was made by packing ∼150 mg of COP-99 in a glass column with a diameter of 4 mm, sandwiched with cotton wool. The total length of the column was 20 cm, and the packed sample length was ∼3 cm. The dye mixture was prepared in a 1:1

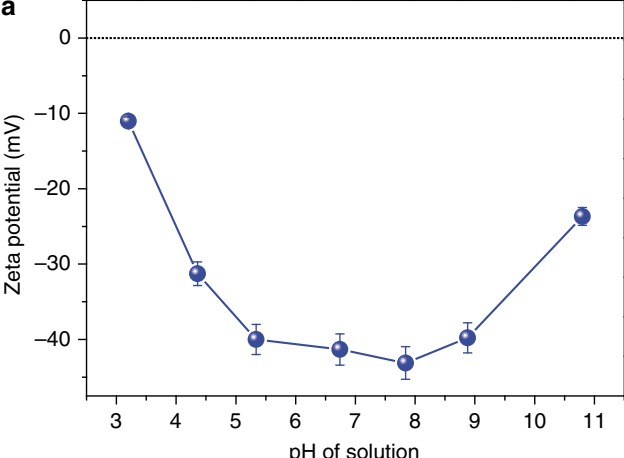

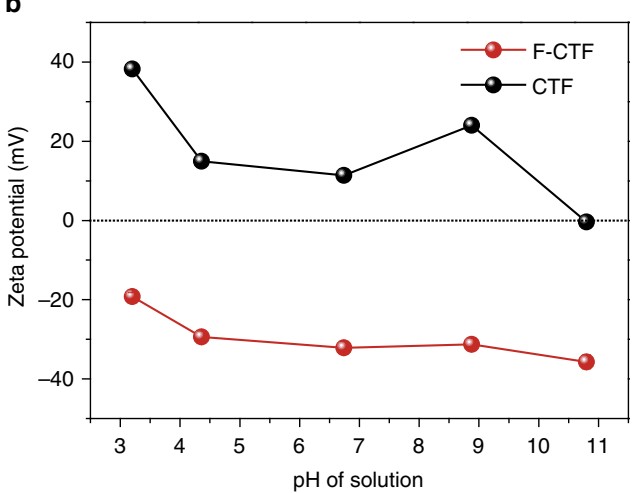

**Figure 6 | Zeta potentials.** pH-dependent zeta potential of (**a**) COP-99 and (**b**) CTF and F-CTF. Error bars are given to represent s.d.

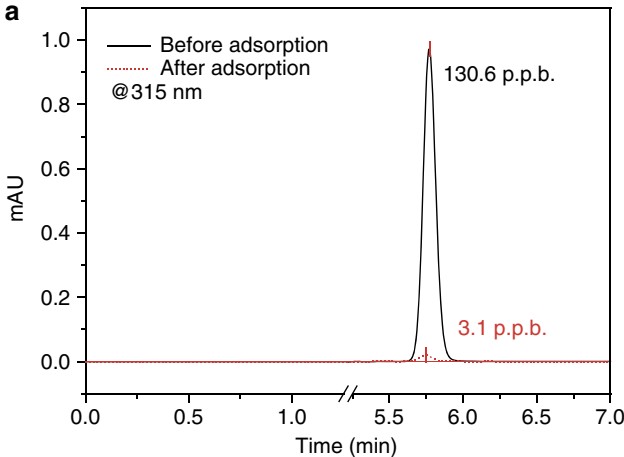

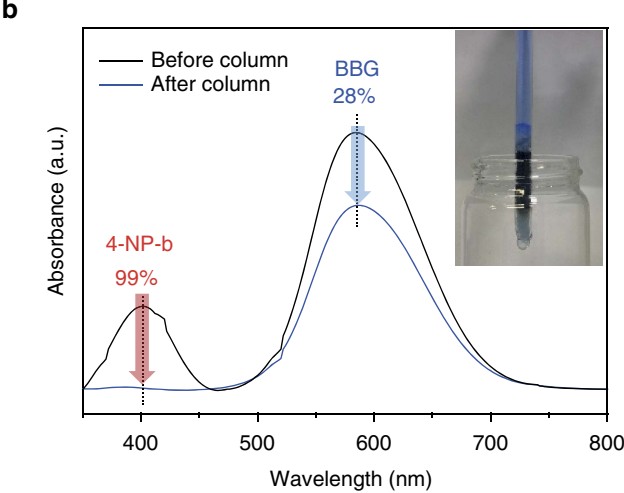

**Figure 7 | Feasibility test of COP-99. (a)** LC spectra of 4-NP-a before and after the treatment with COP-99. **(b)** ultraviolet–visible spectra of mixed dye solution before and after passing through a packed column of COP-99. The initial dye mixture was prepared in a 1:1 v/v ratio of 50 μM BBG and 30 μM 4-NP-b. Inset displays the photograph of the actual column tested in the study.

v/v ratio of 50 μM BBG and 30 μM 4-NP-b, in which the smaller target molecule—4-NP—was in lower concentration to simulate actual conditions. As shown in Fig. 7b, the ultraviolet spectra before and after column treatment exhibited that the dye mixture was clearly separated with a sharp decrease of 4-NP concentration. The concentration change was found to be 28 and 99% for BBG and 4-NP, respectively, indicating the effective separation of the two molecules. The BBG, despite of no uptake in a batch condition, showed a moderate uptake after the column, and it may be attributed to tight packing density of the column. In fact, batch adsorption test with the 4-NP/BBG mixture exhibited that the smaller 4-NP was completely removed from the mixture over 6 h of soaking while larger BBG did not show change in concentration during the treatment (Supplementary Fig. 16c and Supplementary Note 3). The low concentration uptake and column separation test have demonstrated the feasibility of COP-99 for the actual field applications.

## Discussion

We have shown the first charge-specific size-dependent separation of soluble organic species from water and investigated the unprecedented interaction of fluorinated surfaces with charged organic molecules. We designed and synthesized a new fluorinated network polymer, indexed as COP-99 via self-condensation of TFHQ, which exhibits high fluorine content with narrow microporosity. The COP-99 displayed size-selectivity towards charged dye molecules, MB, RDB and BBG, and only MB whose hydrodynamic size is within the accessible pore size distribution of COP-99 is effectively removed from water. COP-99 demonstrated almost no uptake for BPA, a water-soluble molecule with similar dimensions of MB but not charged, confirmed the lack of charge on the guest molecule hinders the adsorption despite size fitting. These findings were validated on other nanoporous materials, namely CTF and F-CTF, having almost identical pore size distributions but different wall functionalities. Fluorinated CTF has shown much higher and faster uptake of small-sized charged molecule owing to the fluorine-cationic site interaction. Our results show a clear fluorine-cation interaction in size-selective adsorption, which can lead to sorbent designs that can be effectively used for size-dependent organic molecule separation from water. In addition, the fluorine-cation interaction can provide great use in mixed matrix membranes with inorganic cations and fluorinated polymers. Based on our findings, we also suggest that fluorinated molecules can be selectively separated (especially from other halogenated organics) by inorganic porous solids with open metal sites[56].

## Methods

**Materials.** TFHQ (97%) and tetrafluoroterephthalonitrile (98%) were purchased from TCI, Japan. Anhydrous potassium carbonate (99.5%), anhydrous zinc chloride (98%), bisphenol A (98%, named BPA) and RDB ($>95\%$, RDB) were from SAMCHUN, South Korea. MB solution (0.05 wt% in $H_2O$, MB), 1,4-dicyanobenzene (98%), 4-NP (99.5%, 4-NP) and BBG were obtained from Sigma-Aldrich, USA. All samples for adsorption test were prepared using de-ionized water (DIW, 18.2 MΩ cm).

**Synthesis of COP-99.** Fluorinated COP-99 was produced via a modified polymer of intrinsic microporosity[47] making procedure. TFHQ (0.5 g) dissolved in 15 ml N,N-dimethylformamide and heated to 80 °C under Ar atmosphere. Anhydrous $K_2CO_3$ (0.25 g) was slowly added to the above solution, and the reaction temperature was increased up to 145 °C with the rate of 5 °C min$^{-1}$. The mixture was vigorously stirred at 145 °C under Ar atmosphere for 24 h. The reaction mixture was cooled down to room temperature, and 100 ml of water was added and stirred for 4 h at room temperature. The obtained product was filtered and thoroughly washed with DIW (100 ml) and acetone (50 ml) until clear filtrate was observed. The resulting brown powder was dried at 120 °C under vacuum for overnight. We note that the $K_2CO_3$ injection speed and the heating rate can affect the pore size distribution, for instance, quick injection of $K_2CO_3$ can broaden the pore size distribution which hinders selective adsorption capability of COP-99 from dye mixtures.

**Synthesis of F-CTF and CTF.** F-CTF and CTF were produced according to procedure reported elsewhere[42,43]. Nitrile monomer and zinc chloride were mixed, packed, evacuated and flame-sealed in borosilicate glass ampoule. The polymers were generated by heating the ampoule at 400 °C for 40 h. Tetrafluoroterephthalonitrile and terephthalonitrile was used for F-CTF and CTF synthesis, respectively, and the molar ratio between nitrile monomer and $ZnCl_2$ was 1. The obtained powder was washed with DIW and further treated with diluted HCl for 12 h to remove residual salt. The resulting black powder was rewashed with DIW and THF, and dried at 150 °C under vacuum for overnight.

**Characterization.** FE-scanning electron microscopy (Field Emission Scanning Electron Microscopy) was performed using a Nova 230. Zeta potential measurements were carried out on an ELS-Z2, Otsukael. XPS (X-ray photoelectron spectroscopy) spectra were obtained using a Thermo VG Scientific Simga Probe system equipped with an Al-Kα X-ray source with an energy resolution of 0.47 eV full-width at half-maximum under ultrahigh vacuum conditions of 10$^{-10}$ Torr. $^{19}$F magic-angle spinning NMR spectrum was acquired in solid-state using an Agilent 400 MHz 54 mm NMR system. Porosity of samples was analysed with Micromeritics Triflex accelerated surface area and porosimetry analyser at 87 K after all the samples were degassed at 150 °C for 6 h under vacuum. The specific surface area of the samples was calculated by BET model, and the pore size distribution was determined by NLDFT approach. Low-pressure gas isotherms ($CO_2$ and $N_2$) were measured using a Micromeritics Triflex system at desired

temperature after degassing samples with the method described above. Thermogravimetric analysis was conducted with a differential thermal gravimetry (DTG)-60A of Shimadzu by heating samples up to 800 °C at a rate of 10 °C min$^{-1}$ under desired atmosphere. X-ray diffraction pattern of samples was acquired from 5 to 80° by using a Rigaku D/MAX-2500 Multi-purpose High power X-ray diffractometer. FT-IR spectra were recorded on KBr disks using a Perkin-Elmer FT-IR spectrometer. Elemental analysis (CHNO) was performed by a sFlash 2000 series of Thermo Scientific. Fluorine content was measured using combustion ion chromatography with Metrohm AQF-100, 881 Compact IC pro. The water contact angle measurement was conducted using a Contact Angle Analyzer Phoenix 300 Plus with the sample powder packed on a glass substrate.

**Selective dye uptake experiment.** All the batch adsorption experiments were conducted at room temperature in the dark condition. Initial dye concentrations were adjusted to be 50 μM and pH of dye solution was controlled to be 8 using 0.1 M NaOH or HCl. In a typical adsorption study, 10 mg of adsorbents (COP-99, F-CTF, CTF and ACN) was dispersed into 10 ml of dye solutions, and the mixture was slowly tumbled on end-over-end (8 rpm). At appropriate time interval, the aliquots were taken from the mixture, and the adsorbents were separated by syringe filter (0.45 μm pore size, Nylon, Whatman). The dye concentration in the solutions was detected using a ultraviolet–visible spectrophotometer (V-570, Jasco) at a wavelength of maximum absorbance (315 nm for 4-NP-a, 405 nm for 4-NP-b, 276 nm for BPA, 664 nm for MB, 554 nm for RDB and 595 nm for BBG). Cyclic experiment was conducted with 4-NP-a of 50 μM concentration. COP-99 was regenerated by being soaked in acetone (1 mg COP-99 per 1 ml acetone) and tumbled for 1 h at 8 r.p.m. The treated COP-99 was filtered and dried at 60 °C for 30 min under air for the cycle experiment. The concentration change of 4-NP-a was analysed by a ultraviolet–visible spectrophotometer as described above. The percentage removal of dyes was calculated as following equation (1):

$$\text{Removal percentage }(\%) = \frac{C_0 - C_t}{C_0} \times 100 \qquad (1)$$

where $C_0$ and $C_t$ are the concentration of dyes at initial and certain time, respectively.

HPLC was performed with an Agilent 1260 HPLC equipped with a Diode-Array Detector ultraviolet detector and a Zorbax Eclipse XDB-C18 column (4.6 × 150 mm, 5 μm particle size). The sample injection volume was 20 μl, and the flow rate was 1.5 ml min$^{-1}$. The mobile phase for HPLC analysis varied depending on the target molecules, that is, methanol 60%: water 40% for NP-b, acetonitrile 30%: water 70% acidified with H$_3$PO$_4$ to pH ~3.5 for MB, and acetonitrile 40%: water 60% for BPA. The intensity of the effluent ultraviolet absorbance was monitored at $\lambda = 335$ nm, 635 nm and 276 nm for NP-b, MB and BPA, respectively. LC calibration curve was created for quantitative analysis using five standard solutions, that is, 0.05, 0.1, 10, 50 and 100 p.p.m. (Supplementary Fig. 17).

For a column separation study, ~150 mg of COP-99 was finely ground by a mortar and packed in a glass column, sandwiched between a cotton wool to prevent a mass loss. The inner diameter of the column was ~4 mm, and total length of the column was 20 cm. The packed sample length in the column was ~3 cm. Dye mixtures for a column separation were prepared in a 1:1 v/v of 50 μM BBG and 30 μM 4-NP-b (4-NP/BBG), and a 1:1 v/v of 50 μM MB and 50 μM BPA (MB/BPA). About 10 ml of dye mixture was passed through the column, and the effluent was collected and analysed by the ultraviolet–visible spectrophotometer to determine the concentration change. The dye mixtures were utilized for batch adsorption tests in which 8 mg of COP-99 was immersed in 8 ml of mixed solution for desired time period.

**Data availability.** MarvinSketch version 15.6.8 was used for the size calculation of organic molecules tested for the adsorption test in this paper, available at: https://www.chemaxon.com/products/marvin/marvinsketch/. The authors declare that the other data supporting the findings of this study are available on request.

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

## Acknowledgements

We acknowledge the financial support by the National Research Foundation of Korea (NRF) grant funded by the Korea government (MSIP; No. NRF-2016R1A2B4011027), (MEST; NRF-2012-C1AAA001-M1A2A2026588), and High Risk High Return Project of the Korea Advanced Institute of Science and Technology (N11160084).

## Author contributions

J.B. carried out all size and charge-dependent studies and produced all materials, H.A.P. developed COP-99, D.T. helped in discussions, C.T.Y. conceived and supervised the study, and wrote the paper with input from all authors.

## Additional information

**Competing financial interests:** The authors declare no competing financial interests.

