## [Peer Review File · Nature Communications]

Reviewers' comments:

Reviewer #1 (Remarks to the Author):

In this manuscript, the authors report the first charge specific size dependent separation of water-soluble organic molecules using a newly designed COP. They discovered the first time that fluorine functionalities could contribute to the selective uptake of charged species over uncharged species of similar size. It is the opinion of this reviewer that this paper is suitable for publication in Nature Communications if the following issues have been addressed.

Page 3: The authors haven't stressed the importance of charge specific or size-dependent separation in removing water soluble organic molecules.

Page 11: "Since there are noticeable plateaus, we suspected that there might be favourable interactions of fluorines with MB" This statement was not very convincing. Please give more explanations on why plateaus might be related to fluorine functionalities. Also, is the data in Figure 4a from experiments done in duplicate or triplicate? Please clarify.

Page 15: How is the removal of 4-NP in acidic/basic form by CTF and F-CTF compared with that of COP-99?

It is the opinion of this reviewer that this manuscript is suitable for publication if these issues are addressed.

Reviewer #2 (Remarks to the Author):

The authors reported a fluorinated porous polymer (COP-99) in the selective adsorption of soluble organic molecules from water. The synthetic methodology of the material has been reported elsewhere (Adv. Mater. 2004, 16, 456-459 et al.), and the fluorine contained porous polymers have also been well-documented (Sci. Rep. 2015, 5, 10155 et al.), thus the novelty of this work mainly relies on the adsorption capability of the porous polymer for water soluble organic compounds. However, the adsorption performances were not quantifiably analyzed and far from comparable with some benchmark materials. Thereby this work is not novel, significant, and competitive enough to be published in Nature Communications. Some additional issues are listed below.

- 1) The concept of charge specific or size-dependent separation is not new in the field of porous materials, even in the context of dye separation from aqueous solutions based on MOFs and porous polymers.
- 2) Many microporous polymers constructed by the similar route have been used for the wastewater treatment (Green Chem. 2015, 17, 5196-5205; Macromolecules, 2015, 48, 5663-5669 et al.)
- 3) Recycling tests should be performed to check the reusability of the material.
- 4) The role of the oxygen species should be discussed, since the hydrophilicity of the material should play an important role in capturing the organic compounds in water.
- 5) UV-vis spectrophotometer is not sensitive enough to detect the organic pollutants in ppb level. Other sophisticated equipment such as HPLC-MS should be used to quantitate the removal efficiency.
- 6) The solubility of the COP-99 in the water should be tested. Based on the connecting mode, the resultant polymer can probably be partially soluble in the solvents.
- 7) Some explanations should be added to address the phenomena observed in the Figure 3b, that is the adsorption speed is slow during the first hour, but much faster in the last hour.
- 8) Some insightful studies should be conducted to understand the interactions between the dye

molecules and the porous polymer material.

Reviewer #3 (Remarks to the Author):

Authors report charge specific size dependent separation of water soluble molecules by fluorinated material. After careful evaluation of the manuscript, will be accepted with major revision. Here are my points that need to be addressed.

1) Size dependent separation in porous materials is a well studied phenomenon as a result I do not see this paper providing any new scientific insights. Most of the articles published on size dependent study focused on larger and smaller organic molecules to separate using porous materials with pore diameter smaller than the bigger molecule. However I would like to suggest the authors to perform similar experiments using para-, ortho- and meta- nitrophenols to demonstrate size and shape behavior with COP-99.

2) Similarly I would recommend the authors to compare the separation behavior with 4-aminophenol vs 4-nitrophenol, Does the hydrogen bonding amino groups influence its adsorption on to the pore walls over the nitro groups. Experiments with these two organic molecules with same size but different hydrogen bonding characteristics will provide an interesting set of results under different pH conditions.

3) In Page 12, COP-99 do adsorb 12% of separation after 3h which is still significant. Can authors explain the adsorption? Does the COP-99 start to expand overtime to accommodate the larger molecule? Performing longer experiments will provide some information. Additionally any other spectroscopic methods to elucidate this mechanism would be an ideal.

4) When submitting for such a high impact paper, the reviewer would like to see column breakthrough experiments with mixed organic molecules in the same solution instead of performing experiments individually. I highly recommend these experiments with variable concentrations. For example, trace amounts of 4 nitro phenol in low concentrations vs higher concentrations of larger organic molecules.

Reviewer #1 (Remarks to the Author):

In this manuscript, the authors report the first charge specific size dependent separation of water-soluble organic molecules using a newly designed COP. They discovered the first time that fluorine functionalities could contribute to the selective uptake of charged species over uncharged species of similar size. It is the opinion of this reviewer that this paper is suitable for publication in Nature Communications if the following issues have been addressed.

We are grateful to the reviewer for taking time to evaluate and acknowledging the importance of our work.

(Comment 1) (Page 3) The authors haven't stressed the importance of charge specific or size-dependent separation in removing water soluble organic molecules.

(Answer 1)

We appreciate the reviewer for the important suggestion. We now modify the main text to better emphasize the significance of the size/charge-specific interaction:

(Page 3) "... with fine dimensions (especially sub nm) is very challenging^{30,31}. Common sieving methodologies that are used by the gas separation media (Nature 495, 2013, 80-84) may not always work since the presence of strong intermolecular forces in aqueous media requires new approaches such as charge specific size dependent separations. Despite a few attempts for the removal of organic molecules using nanoporous ..."

(Comment 2) (Page 11) "Since there are noticeable plateaus, we suspected that there might be favourable interactions of fluorines with MB" This statement was not very convincing. Please give more explanations on why plateaus might be related to fluorine functionalities. Also, is the data in Figure 4a from experiments done in duplicate or triplicate? Please clarify.

(Answer 2)

We thank the reviewer for his/her insightful comment. We kindly note that all the batch adsorption tests shown in Figure 4a were done in duplicates, and we now update Figure 4a with error bars of standard deviation from two different batches:

Figure 4 | Size selective separation of water-soluble substrates by COP-99. (a) Change in dye concentrations over time after being treated with COP-99 in terms of absorbance relative to initial absorbance (C/C_0). Initial concentration (C_0) of all the dyes was adjusted to be $50 \mu\text{M}$. 4-Nitrophenol was tested both in acidic (4NP-a, $\text{pH} = 4$) and basic (4NP-b, $\text{pH} = 9$) conditions. **Error bars reflect standard deviation from duplicates.** (b) Surface area distribution with respect to pore size, and its relation toward minimum van der Waals size of dye molecules.

More so than the standard deviation, we found a consistent adsorption plateau in the first 30 min of MB adsorption. This is in line with a study by Herm et al. (*Science*, 2013, 340, 960-964) where hexane isomers showed a “stepwise adsorption” due to the unique structure and alignment of triangular pores:

➤ Stepwise adsorption of hexane isomers because of the rolling mechanism (left) due to the size and shape of the guest molecule

Herm et al., 2013, Science, 960-964

[Quotation from Herm et al.] “Figure 2 ... Snapshots of the hexane isomers within the channels of

Fe₂(BDP)₃ for a loading of four molecules per unit cell at 160°C, as observed in CBMC simulations.” And “At 160°C, the two dibranched hexane isomers **display stepwise adsorption** with an inflection point near 100 mbar. The stepwise uptake of alkanes has been observed previously with cyclohexane (30) and n-hexane (31) adsorption, and can be explained with entropic arguments, as supported by calorimetric data (32).…”

In our system, we suspect that adsorption plateaus are attributed to favourable electrostatic interactions between fluorine on COP-99 and ammonium cation site of MB molecule. If the pore size were the only limiting factor in MB diffusion, linear-type of adsorption isotherm would appear, as the guest molecule would've not favourably interacted with the surface of COP-99. But in our case, as the MB molecules approach, they get adsorbed on the most available sites (i.e., fluorines on the surface and outer rims of the grains). After the first 30 min, the surface active sites are fully occupied with MB molecules making the adsorption plateau more prominent than in the work by Herm et al. This is because of the favourable, stronger interaction of fluorines with cations as opposed to the van der Waals based *physisorption* of hexanes in the latter. Once enough substrate build-up is observed, the stepwise adsorption takes place by shuffling guest molecules to the interior. This, in return, provides a new push towards higher uptake only before getting saturated again.

(Page 11) In a pore limited diffusion scenario, kinetics is equally hindered (hence linear curve with a constant slope) for all the substrate species provided that the wall surfaces do not interact favourably with the incoming guests. **The favourable interaction between adsorbate and adsorbent should, therefore, affect the adsorption kinetics. The two concepts are best showcased in Long et al. (*Science* 340, 2013, 960-964), where hexane isomers were shuttled inside a metal organic framework in a stepwise fashion, creating pseudo saturations, akin to monolayer – multilayer coverage switching in hierarchical porous systems.** Since there also are noticeable plateaus in MB adsorption, we suspected that there might be favourable interactions of fluorines with MB.

We truly thank the reviewer for his/her contribution.

(**Comment 3**) (Page 15) *How is the removal of 4-NP in acidic/basic form by CTF and F-CTF compared with that of COP-99?*

(**Answer 3**)

We are grateful to the reviewer for his/her suggestion. We now tested the removal of 4-NP molecule under acidic and basic conditions with CTF and F-CTF structures. In identical experimental conditions with COP-99, CTF and F-CTF exhibited faster uptake of 4-NP-a and 4-NP-b than COP-99, mainly due to the larger pore size and surface area. Both CTF and F-CTF removed 99 % of 4-NP-a within 4 min and 1 min, respectively, while CTF and F-CTF captured about 99% and 95 % of 4-NP-b for 12 min of adsorption, respectively. Compared to CTF, F-CTF showed enhanced uptake of 4-NP, removing 99 % of 4-NP-a in the first 1 min of adsorption, which should be due to the fluorine functionalities on F-CTF. However, we note that the pore size of both CTF and F-CTF is large enough to capture small 4-NP molecules, thus 4-NP molecules can enter the pores of CTF and F-CTF without interacting with the surface functionalities of the CTF and F-CTF, as shown in Figure 4a from COP-99.

We now include Figure S13 and the description related to 4-NP uptake by using CTF and F-CTF in the main text as follows:

(Page 17) When the size of sorbate molecules is small enough, i.e. 4-NP-a and 4-NP-b, both CTF and F-CTF readily capture the dyes (Supplementary Fig. 13). CTF and F-CTF exhibited much faster uptake of 4-NP-a and 4-NP-b than COP-99, mainly due to the larger pore size and higher surface area. Both CTF and F-CTF removed 99 % of 4-NP-a within 4 min and 1 min, while CTF and F-CTF captured about 99% and 95 % of 4-NP-b for 12 min of adsorption, respectively. The uptake is also noticeably faster in F-CTF, mainly due to the interaction of fluorines with charged molecules.

Figure S13. Change in 4-NP concentrations over time after being treated with (a) CTF and (b) F-CTF in terms of absorbance relative to initial absorbance (C/C_0). Initial concentration (C_0) of the dyes was adjusted to be 50 μ M. 4-Nitrophenol was tested both in acidic (4NP-a, pH = 4) and basic

(4NP-b, pH = 9) conditions.

It is the opinion of this reviewer that this manuscript is suitable for publication if these issues are addressed.

We sincerely thank the reviewer for his/her contribution.

Reviewer #2 (Remarks to the Author):

The authors reported a fluorinated porous polymer (COP-99) in the selective adsorption of soluble organic molecules from water.

We thank the reviewer for taking time to review and contribute to our work.

The synthetic methodology of the material has been reported elsewhere (Adv. Mater. 2004, 16, 456-459 et al.), and the fluorine contained porous polymers have also been well-documented (Sci. Rep. 2015, 5, 10155 et al.), thus the novelty of this work mainly relies on the adsorption capability of the porous polymer for water soluble organic compounds.

We agree with the reviewer that the synthesis methodology is previously reported. And we duly reference the paper in the manuscript as reference 47. There are, however, a vast number of papers that use the same method to make highly porous/functional systems. One last example is the letter in *Nature* (190-194, 2016) by Dichtel et al (also cited as ref 37) where cyclodextrins are attached to PIM (Polymers of Intrinsic Microporosity) monomer through the same nucleophilic reaction. We believe that the reaction works well and suitable to make new and exciting materials. We are also advancing the methodology (although not as significant as our main finding) by carrying out the first self-condensation version of the PIM formation reactions (and the most affordable). We apologize for not citing the *Scientific Reports* (2015, 5, 10155) paper (and now added in the revised version as reference 38) but since the findings reported there is in line with the letter in *Nature*, we have had an unintentional oversight to give due credit. We are sorry for that and now reflect the findings of that paper in the revised main text as follows:

(Page 4) "... similar to the common activated carbons. Deng et al. also reported superhydrophobic fluorine containing, microporous (< 2 nm) polymers with high uptake capacities for organic dyes and lead (*Sci. Rep.* 5, 2015, 10155). Perhaps, lack of ..."

That being said, we would like to remind the reviewer that our finding is not about how high or fast we can remove organic contaminants. Our discovery is that fluorine shows particular interest in cationic species and when coupled with size sieving properties it is shown to separate charged species. This has never shown before and we believe it is worth publishing in a high impact journal like *Nature Communications* since a wide readership will benefit from our discovery.

However, the adsorption performances were not quantifiably analyzed and far from comparable with some benchmark materials. Thereby this work is not novel, significant, and competitive enough to be published in Nature Communications. Some additional issues are listed below.

With all due respect, we believe we have given careful attention to our analysis. We'll be very happy if the reviewer can explain further why our adsorption results were not "quantifiably analysed". We think we adhered to the standard analytical protocols with precision equipment and repetitive, reproducible experimentation. And since there is no evidence of the fluorine – cation interaction prior to our study (certainly no charge specific size dependent separation), we don't know how we can benchmark our materials. We'd be happy if the reviewer can suggest ways to do this.

(Comment 1) The concept of charge specific or size-dependent separation is not new in the field of porous materials, even in the context of dye separation from aqueous solutions based on MOFs and porous polymers.

(Answer 1)

We agree with the reviewer that separation of organic molecules is reported in a number of reports (*Green Chem.* 2015, 17, 5196-5205; *Macromolecules*, 2015, 48, 5663-5669, *Sci. Rep.* 2015, 5, 7910, *Nature* 2016, 190-194) but only few focused on size dependence (*Langmuir* 2006, 22, 4225, *Chem. Mater.* 2015, 27, 3207, *Angew. Chem. Int. Ed.* 2015, 54, 12748-12752). When it comes to charge specific size dependence, there is no previous work. In addition, in all size dependent separation studies all the substrate molecules were charged (either anionic or cationic). We'd be happy to consider any suggestions. And again, we report a new feature of fluorine functionality in porous polymers that can facilitate the separations by favourable charge-fluorine interactions.

(Comment 2) Many microporous polymers constructed by the similar route have been used for the wastewater treatment (Green Chem. 2015, 17, 5196-5205; Macromolecules, 2015, 48, 5663-5669 et al.)

(Answer 2)

We are grateful to the reviewer for his/her comment on existing literature. We are aware of the reports the reviewer is highlighting. Instead of citing all the literature on water treatment by microporous polymers (a thorough review would be a good contribution) we resorted to highlight a

few that are as close to our findings as possible. Even among all that literature the reports by Deng et al. (*Sci. Rep.* 2015, 5, 10155) and Dichtel et al. (*Nature* 2016, 190-194) are the only ones that use fluorinated porous polymers for water treatment. And they focused an all-contaminant removal strategy (similar to activated carbons) based on superhydrophobicity and kinetics, both of which are not what we focus here. We discovered fluorine-cation interaction and showed that one can use it for charge specific size dependent separation. Our material, COP-99 is easily synthesized in bulk quantities and can be used in removal applications (though this was also not our primary focus).

(Comment 3) Recycling tests should be performed to check the reusability of the material.

(Answer 3)

We are thankful to the reviewer and per his/her request, we carried out regeneration test of COP-99. 4-NP-a was chosen as a target sorbate molecule, as it quickly reached the adsorption equilibrium in a short time period (~ 30 min). COP-99 was regenerated by soaking in acetone (1 mg COP-99 / 1 ml acetone) and tumbled at 8 rpm for an hour. After being dried at 60 °C for 30 min under air, the COP-99 was used for another adsorption experiment. Each cycle was carried out for 30 min, and the concentration change was analysed every 10 min. COP-99 showed excellent 4-NP-a uptake in a repeated cycle, resulting in negligible capacity loss even after 6th cycle. We now include Figure S12 in the supplementary information, and describe experimental details and reusability of COP-99 in the manuscript as follows:

Figure S12. Removal efficiency of 4-NP-a on COP-99 in six successive cycles of adsorption-desorption. Initial concentration of 4-NP-a was adjusted to be 50 μ M and pH of all the samples was

controlled to be 4.

(Page 13) The small-sized 4-NP molecule also exhibited reversible adsorption-desorption behaviour on COP-99. Throughout a repeated cycle, COP-99 showed excellent uptake of 4-NP-a with negligible loss of removal efficiency after 6th cycle (Supplementary Fig. 12).

(Page 23) Cyclic experiment was conducted with 4-NP-a of 50 μ M concentration. COP-99 was regenerated by being soaked in acetone (1 mg COP-99 per 1 ml acetone) and tumbled for 1 h at 8 rpm. The treated COP-99 was filtered and dried at 60 °C for 30 min under air for the cycle experiment. The concentration change of 4-NP-a was analysed by a UV-vis spectrophotometer as described above.

We thank the reviewer for his/her insightful suggestion to contribute to our work.

(Comment 4) The role of the oxygen species should be discussed, since the hydrophilicity of the material should play an important role in capturing the organic compounds in water.

(Answer 4)

This is a very good advice. As the reviewer pointed out, there were oxygenated species existing in COP-99 verified from XPS and FTIR analysis. In particular, the unreacted hydroxyl units in COP-99 may have contributed to the selective adsorption. We, thus, carried out FTIR analysis after the adsorption of MB molecule to confirm which functional groups of COP-99 affect dominantly on the selective adsorption behaviour. As shown in the figure below, COP-99 showed insignificant change in FTIR spectra after the MB adsorption owing to the physisorptive interaction between COP-99 and MB molecule. Nonetheless, there was one minor peak disappearing after the MB adsorption, which is at the position for F-phenyl ring bending mode at 740 cm^{-1} (*Langmuir* 1998, 14, 1227; *Spectrochimica Acta* 1968, 24, 1999). This may be attributed to the hindrance of C-F bond angle owing to the adsorption of MB molecules on the fluorine. Therefore, we conclude that the hydroxyl unit on COP-99 does not show distinct change in IR spectrum, implying that the role of the oxygenated species in COP-99 was rather minor for the adsorption of organic molecules.

Figure S3b. Change in FTIR spectrum after the adsorption of MB molecule.

In order to further study the comment by the reviewer, we also conducted water contact angle (CA) measurement to check the hydrophilicity of COP-99. As seen in the Figure S9b, COP-99 exhibits surface hydrophobicity with a water contact angle of 142° . Such a hydrophobic nature of COP-99 results from the high amounts of fluorine units in the COP-99. And if added to water, COP-99 floats on the surface (Figure S9c).

Figure S9. (b) Contact angle for a water droplet on the surface of COP-99. (c) Photograph of COP-99 in water. Owing to the hydrophobicity, COP-99 floats on the surface of water.

We now include Figure S9 in the supplementary information, and updated the main text accordingly to describe the findings in IR spectrum and hydrophobicity of COP-99 as follows:

(Page 9) As it is expected, plenty of fluorines in COP-99 make the structure hydrophobic. Water contact angle (CA) measurement shows that COP-99 exhibits surface hydrophobicity with a water contact angle of 142° (Supplementary Fig. 9b). If added to water, COP-99 floats on the surface (Supplementary Fig. 9c). Hydrophobic nature of polymeric adsorbent is also known to be helpful for adsorption of organic molecules (*Sci. Rep.* 5, 2015, 10155).

(Page 11) The specific interaction between fluorine on COP-99 and MB molecule was confirmed by FTIR spectrum (Supplementary Fig. 3b). After the MB adsorption, COP-99 showed insignificant change in FTIR spectrum due to the weak electrostatic interaction between COP-99 and MB molecule. Nonetheless, there was one minor peak disappearing through the MB adsorption, which is at the position for F-phenyl ring bending mode at 740 cm⁻¹. This may be attributed to the hindrance of the C-F bond angle as the MB molecule favourably binds on the fluorine of COP-99.

We appreciate the reviewer for his/her suggestion to clarify our result.

(Comment 5) UV-vis spectrophotometer is not sensitive enough to detect the organic pollutants in ppb level. Other sophisticated equipment such as HPLC-MS should be used to quantitate the removal efficiency.

(Answer 5)

We thank the reviewer for the insightful comment. As the reviewer might agree, UV-vis spectroscopy is one of the simplest and fastest analytic tools with reliable accuracy. This is why we mainly used UV-Vis measurement for the convenience of experiments. But complying with the reviewer's comment, we have now carried out HPLC analysis to monitor concentration change in ppb level. 4-NP-a was tested as a target molecule and the initial concentration was adjusted to be about 100 ppb (~ 0.72 μM). The mobile phase for HPLC analysis consisted of 65 % water/35 % acetonitrile, acidified with H₃PO₄ to pH ~3.5. LC calibration curve was created for quantitative analysis using five standard concentrations of 4-NP-a, i.e. 0.05 ppm, 0.1 ppm, 10 ppm, 50 ppm, and 100 ppm. The intensity of the effluent UV absorbance was monitored at λ = 315 nm.

The LC spectrum of initial 4-NP-a solution showed a single sharp peak at 5.77 min of retention time, and after being treated with COP-99 for 1 h, the peak significantly diminished as shown at 5.75 min of retention time. From the analysis with the LC calibration curve, the actual concentration of the initial and the treated solution was found to be 130.6 ppb and 3.1 ppb, respectively, indicating the removal efficiency was about 98 %. This result agrees well with the

removal efficiency of 4-NP-a tested under higher concentration with UV-Vis analysis. The HPLC study also concluded that COP-99 could clean organic pollutants even at very low concentrations. We thank the reviewer for this valuable contribution.

We now include the HPLC discussion and experimental details in the main text and supplementary information as follows:

(Page 19) First, we carried out an adsorption test of 4-NP in ppb level and traced the change in concentration using HPLC, which enables us to detect 4-NPs in a more quantitative manner at these concentrations. The initial concentration of 4-NP was adjusted to be about 100 ppb, and the adsorption was carried out under a mild acidic condition for 1 h (4-NP-a). The LC spectrum of initial 4-NP-a solution showed a single sharp peak at 5.77 min of retention time, and after being treated with COP-99 for 1 h, the peak significantly diminished as shown at 5.75 min of retention time (Fig. 7a). From the analysis with the LC calibration curve, the actual concentrations of the initial and the treated solution were found to be 130.6 ppb and 3.1 ppb, respectively, indicating the removal efficiency was about 98 %. This result agrees with the removal efficiency of 4-NP-a tested under higher concentration with UV-vis spectrum analysis.

(Page 23) High performance liquid chromatography (HPLC) was performed with an Agilent 1260 HPLC equipped with a DAD (Diode-Array Detector) UV detector and a Zorbax Eclipse XDB-C18 column (4.6 x 150 mm, 5 μ m particle size). The sample injection volume was 20 μ l, and the flow rate was 1.5 ml/min. The mobile phase consisted of 65 % water/35 % acetonitrile, acidified with H₃PO₄ to pH ~3.5, and the intensity of the effluent UV absorbance was monitored at $\lambda = 315$ nm. LC calibration curve was created for quantitative analysis using five standard solutions, i.e. 0.05 ppm, 0.1 ppm, 10 ppm, 50 ppm, and 100 ppm.

Figure 7a. LC spectra of 4-NP-a before and after the treatment with COP-99.

Figure S14. HPLC calibration curve of 4-NP-a from five different concentrations of 4-NP-a: 0.05 ppm, 0.1 ppm, 10 ppm, 50 ppm, and 100 ppm.

We are grateful to the reviewer for his/her contribution.

(Comment 6) The solubility of the COP-99 in the water should be tested. Based on the connecting mode, the resultant polymer can probably be partially soluble in the solvents.

(Answer 6)

From the contact angle measurements and water testing (Figure S9), we found out that COP-99 is hydrophobic and insoluble in water. We have also tested solubility of COP-99 in different organic solvents (Figure S9a). COP-99 was not soluble in most of common organic solvents even after sonication for 1 min.

A typical linear PIM structure is known to be solubilized in common organic solvents. COP-99, despite the similar synthetic condition, is a network polymer by the self-condensation of a single monomer, thus it is not readily soluble in solvents. As the COP-99 targets water purification, a rigid, insoluble structure is a necessity. As the reviewer suggested, we now clarify the solubility issue of COP-99 by including a figure and description as follows:

Figure S9. (a) Solubility test of COP-99 in common solvents. Images were taken after a sonication for 1 min at 50 °C.

(Page 5) Due to the highly cross-linked network structure, COP-99 was insoluble in common solvents (Supplementary Fig. 9a).

(Comment 7) Some explanations should be added to address the phenomena observed in the Figure 3b, that is the adsorption speed is slow during the first hour, but much faster in the last hour.

(Answer 7)

We're glad that the reviewer pointed out an important finding. There is an adsorption plateau in the first 30 min of MB adsorption, where the adsorption speed was slow during at the initial stage, and became faster after sometime. These findings are similar, in principle, with a study by Herm et al. (*Science*, 2013, 960-964) where hexane isomers showed a “stepwise adsorption” due to the unique structure and alignment of triangular pores:

- Stepwise adsorption of hexane isomers because of the rolling mechanism (left) due to the size and shape of the guest molecule

Herm et al., 2013, Science, 960-964

[Quotation from Herm et al.] “Figure 2 ... Snapshots of the hexane isomers within the channels of $\text{Fe}_2(\text{BDP})_3$ for a loading of four molecules per unit cell at 160°C , as observed in CBMC simulations.” And “At 160°C , the two dibranched hexane isomers **display stepwise adsorption** with an inflection point near 100 mbar. The stepwise uptake of alkanes has been observed previously with cyclohexane (30) and n-hexane (31) adsorption, and can be explained with entropic arguments, as supported by calorimetric data (32)...”

In our system, we suspect that adsorption plateaus are attributed to favourable electrostatic interactions between fluorine on COP-99 and ammonium cation site of MB molecule. If the pore size were the only limiting factor in MB diffusion, linear-type of adsorption isotherm would appear, as the guest molecule would've not favourably interacted with the surface of COP-99. But in our case, as the MB molecules approach, they get adsorbed on the most available sites (i.e., fluorines on the surface and outer rims of the grains). After the first 30 min, the surface active sites are fully occupied with MB molecules making the adsorption plateau more prominent than in the work by Herm et al. This is because of the favourable, stronger interaction of fluorines with cations as opposed to the van der Waals based *physisorption* of hexanes in the latter. Once enough substrate build-up is observed, the stepwise adsorption takes place by shuffling guest molecules to the interior. This, in return, provides a new push towards higher uptake only before getting saturated again. We, thus, conclude that the slight adsorption plateau found in MB adsorption may result from the specific (i.e., favourable) interaction between electronegative fluorine and cationic MB molecule. We now include the following description for clarify the MB adsorption behaviour of

COP-99:

(Page 11) In a pore limited diffusion scenario, kinetics is equally hindered (hence linear curve with a constant slope) for all the substrate species provided that the wall surfaces do not interact favourably with the incoming guests. **The favourable interaction between adsorbate and adsorbent should, therefore, affect the adsorption kinetics. The two concepts are best showcased in Long et al. (Science, 2013, 340, 960-964), where hexane isomers were shuttled inside a metal organic framework in a stepwise fashion, creating pseudo saturations, akin to monolayer – multilayer coverage switching in hierarchical porous systems.** Since there also are noticeable plateaus in MB adsorption, we suspected that there might be favourable interactions of fluorines with MB.

(Comment 8) Some insightful studies should be conducted to understand the interactions between the dye molecules and the porous polymer material.

(Answer 8)

We thank the reviewer for his/her suggestion. We have now looked into verifying the interaction between the dye molecule and COP-99, and for that purpose we analysed the spectroscopic difference on COP-99 before and after the adsorption of MB molecule. First of all, in order to better understand the difference in IR spectra by the adsorption, we updated IR spectrum of Figure S3 (now Figure S3a) with the spectra of related monomers for a comparison. FTIR spectrum of COP-99 was also re-taken on KBr disk (previously taken by ATR system) for better resolution. As displayed in the updated figure, COP-99 exhibited free O-H stretching vibration and H-bonded – OH vibration at 3670 cm^{-1} and 3300 cm^{-1} , respectively. The vibration at 1030 cm^{-1} is analysed by the comparison with FTIR spectra of two monomers. Hexafluorobenzene exhibited a strong C-F stretching at 1010 cm^{-1} region, while tetrafluorohydroquinone showed a vibration at 960 cm^{-1} coming from C-O stretching vibration. As the C-F and C-O vibrations are closely located even in monomers, and the strong peak at 1030 cm^{-1} on COP-99 is originated from the combination of C-F and C-O stretching vibrations. The weak vibration at around 740 cm^{-1} results from F-phenyl ring bending vibration (*Langmuir* **1998**, 14, 1227; *Spectrochimica Acta* **1968**, 24, 1999). Interestingly, after the adsorption of MB molecule on COP-99, there was disappearance of the bending vibration at 740 cm^{-1} (Figure S3b). This implies that the specific binding of MB molecule on fluorine of COP-99 affects to the C-F bond angle, leading to the weakening of the C-F deformation vibration. We could not observe the change in 1030 cm^{-1} region after the MB adsorption, as the peak consisted of both C-F and C-O vibrations.

Figure S3. (a) FTIR spectra of COP-99 and related monomer units. (b) Change in FTIR spectrum after the adsorption of MB molecule.

We now update Figure S3 in the supplementary information. The discussion on FTIR spectra after the MB adsorption was added in page 11.

(Page 11) The specific interaction between fluorine on COP-99 and MB molecule was confirmed by FTIR spectrum (Supplementary Fig. 3b). After the MB adsorption, COP-99 showed insignificant change in FTIR spectrum due to the weak electrostatic interaction between COP-99 and MB molecule. Nonetheless, there was one minor peak disappearing through the MB adsorption, which is at the position for F-phenyl ring bending mode at 740 cm^{-1} . This may be attributed to the hindrance of the C-F bond angle as the MB molecule favourably binds on the fluorine of COP-99.

We thank the reviewer for taking time to contribute our study.

Reviewer #3 (Remarks to the Author):

Authors report charge specific size dependent separation of water soluble molecules by fluorinated material. After careful evaluation of the manuscript, will be accepted with major revision. Here are my points that need to be addressed.

We thank the reviewer for valuable comments and appreciation of our work.

(Comment 1) Size dependent separation in porous materials is well studied phenomenon as result i do not see this paper providing any new scientific insights. Most of the articles published on size dependent study focused on larger and smaller organic molecules to separate using porous materials with pore diameter smaller than the bigger molecule. However i would like to suggest the authors to perform similar experiments using para-, ortho- and meta- nitrophenols to demonstrate size and shape behavior with COP-99.

(Answer 1)

We thank the reviewer for his/her comment. We agree with the reviewer that separation of organic molecules is reported in a number of reports (*Green Chem.* 2015, 17, 5196-5205; *Macromolecules*, 2015, 48, 5663-5669, *Sci. Rep.* 2015, 5, 7910, *Nature* 2016, 190-194) but only few focused on size dependence (*Langmuir* 2006, 22, 4225, *Chem. Mater.* 2015, 27, 3207, *Angew Chem Int Ed.* 2015, 54, 12748-12752). When it comes to charge specific size dependence, there is no previous work. In addition, in all size dependent separation studies all the substrate molecules were charged (either anionic or cationic). And again, we report a new feature of fluorine functionality in porous polymers that can facilitate the separations by favourable charge-fluorine interactions.

As the reviewer suggested, and we agree that despite the size fitting, there might be differences in adsorption behaviour when the size and shape of molecules changes. Based on the recommendation, we now conducted adsorption tests of nitrophenol (NP) isomers under acidic conditions, and the concentrations of NPs were controlled to be at 100 μM . The initial concentration was adjusted a bit higher than the original experiment shown in Figure 4 (50 μM), since the size of NP is too small and there is a chance of not seeing any difference among NP isomers in low concentrations. From the size calculations using *MarvinSketch*, theoretical maximum sizes of 4-, 3-, 2-nitrophenol were found to be 0.95 nm, 0.94 nm, and 0.88 nm, respectively. As shown the Figure S11, all three NP molecules were completely removed within 1 h, despite the difference in adsorption kinetics at the first 10 min of adsorption. 2-NP-a exhibited 92 % of removal efficiency while 3- and 4-NP-a showed about 81 % of removal efficiency at the 10 min

of interaction. Interestingly, the smaller 2-NP-a was removed a little bit faster, and 3-NP-a and 4-NP-a having similar size showed nearly identical adsorption isotherms. These results agree with the adsorption studies of larger dyes, MB, RDB, and BBG. Although the chemical nature is very similar, only smaller molecules (than pore opening) were able to penetrate into COP-99. When the size of molecule is much smaller than the pore of COP-99, the smaller molecule is still captured faster, which is a direct evidence of size-dependent adsorption capability of COP-99.

We now include Figure S11 in the supplementary information and description of NP isomer adsorption in the main text as follows:

Figure S11. Change in NP isomer concentrations over time after being treated with COP-99 in terms of absorbance relative to initial absorbance (C/C_0). Initial concentration (C_0) of all the dyes was adjusted to be 100 μ M, and the adsorption was conducted in mild acid condition (pH \sim 3.8). Inset displays the maximum van der Waals diameter of NP isomers calculated by *MarvinSketch*.

(Page 13) The smaller molecule, however, was always taken faster owing to the size-dependent adsorption capability of COP-99. For instance, NP isomers (2-NP and 3-NP), having a slight size difference with 4-NP, also showed a complete removal in the given time. Among three, the 2-NP with smallest size among three, exhibited the fastest adsorption kinetics (Supplementary Fig. 11).

We are grateful to the reviewer for his/her contribution.

(Comment 2) Similarly i would recommend the authors to compare the separation behavior with 4-aminophenol vs 4-nitrophenol, Does the hydrogen bonding amino groups influences its adsorption

on to the pore walls over the nitro groups. Experiments with these two organic molecules with same size but different hydrogen bonding characteristics will provide an interesting set of results under different pH conditions.

(Answer 2)

We thank the reviewer for the helpful suggestions. Per the reviewer's suggestion, we first tested 4-aminophenol (4-AP) uptake using COP-99 under acidic conditions. The theoretical maximum size of 4-AP is 0.898 nm, which is slightly smaller than that of 4-NP. In the mild acidic conditions, we expect the amine group of 4-AP to be partially protonated so that 4-AP could show better adsorption than 4-NP. In the adsorption isotherms, however, the 4-AP showed slower adsorption kinetics than 4-NP with less removal efficiency in the given interaction time of 1 h. We think – among other considerations- this is originated from the fact that 4-AP is known to be decomposing under air and light, and the solubility in water is much lesser than that of 4-NP (10g/L at 20°C for 4-NP, 1.6g/L at 25 °C for 4-AP) (*Hawley's Condensed Chemical Dictionary 15th Edition*. John Wiley & Sons, Inc. New York, 2007, p. 62). Actually, the standard solution of 4-AP in a concentration of 10 mM notably formed precipitates over time with significant colour change into dark brown, implying the instability of 4-AP in water for extended times. We, therefore, think a direct comparison between 4-AP and 4-NP will also be dependent on their chemical behaviour in aqueous solutions.

Figure. Comparison of 4-NP-a and 4-AP-a uptake by COP-99 in terms of absorbance relative to initial absorbance (C/C_0). Initial concentration of 4-NP and 4-AP was adjusted to be 100 μ M, and the adsorption was conducted in mild acid condition (pH \sim 3.8).

Taking into account the idea behind the reviewer's suggestion, we looked for better

substrates for testing the hydrogen-bonding effect on the adsorption. The m-phenylenediamine (m-PD) is chosen as a guest molecule since it has a similar size with NP with two dangling amino groups that can do better hydrogen bonding with COP-99. The calculated maximum size of m-PD was 0.885 nm, which is close to the size of 2-NP. Thus, the adsorption behaviour of m-PD was compared to that of 2-NP for minimizing the size effect in adsorption (Figure S15).

Figure S15. Change in m-PD concentrations over time after being treated with COP-99 in terms of absorbance relative to initial absorbance (C/C_0). Initial concentration (C_0) of all the dyes was adjusted to be 100 μM . The m-PD was tested both in acidic (m-PD-a, pH = 4) and basic (m-PD-b, pH = 10) conditions. The concentration was analysed using a UV-vis spectrophotometer at a wavelength of maximum absorbance (270 nm for m-PD-a and 220 nm for m-PD-b) (*J. Environ. Sci. Heal. A* 2002, 37, 1841). Inset shows molecular structure of m-PD.

As shown in the Figure S15, m-PD was more stable than 4-AP, and both m-PD in acidic and basic conditions was completely removed after 1 h of adsorption. Owing to the protonation of amino group, m-PD-a was shown a bit faster adsorption behaviour than the m-PD-b counterpart, exhibiting 72 % and 60 % at the first 10 min of interaction with COP-99, respectively. Compared to the 2-NP-a, however, the adsorption kinetics of m-PD was rather slower, implying the hydrogen-bonding effect was not as dominant as expected.

In addition to these new experiments, we would like to highlight the behaviour of the BPA (Bisphenol A) adsorption to consider hydrogen-bonding effect in the selective adsorption. As displayed in Figure 4, MB and BPA have very close molecular size, however, the fluorine-charge interaction prompted faster uptake of MB compared to BPA. The BPA showed about 12 % of

removal efficiency for 3 h, and this moderate capacity may be from a number of factors that include (1) sieving, (2) hydrogen bonding between the hydroxyls of BPA and the fluorines on COP-99, and (3) adsorption on the grain surface cavities. Based on the BPA adsorption capacity of 12 %, we assume that the possible hydrogen bonding effect was not predominant than the fluorine-charge interaction.

We now include Figure S15 in the supplementary information and description of m-PD adsorption in the main text as follows:

(Page 13) One might consider that the selective uptake of the adsorbates could originate from hydrogen bonding between the acidic protons of the guest molecules and the fluorines of COP-99. In order to study the hydrogen bonding effect in the adsorption, we tested m-phenylenediamine (m-PD) with two amino groups. The m-PD has a theoretical maximum size of 0.885 nm, close to the 2-NP, so that the adsorption of m-PD was compared to that of 2-NP. Because of its small size (compared to the pore openings), the m-PD was completely removed after 1 h of adsorption in both acidic and basic conditions (Supplementary Fig. 15). Owing to the protonation of amino group, m-PD-a showed a bit faster adsorption behaviour than the m-PD-b counterpart, exhibiting 72 % and 60 % at the first 10 min of interaction with COP-99, respectively. Compared to the 2-NP-a, however, the adsorption kinetics of m-PD was slower, implying that the hydrogen-bonding effect was not as dominant as expected since there is more hydrogen bonding contribution in m-PD. This is in line with the low uptake of BPA using COP-99, where the possible hydrogen bonding on two hydroxyl units does not dramatically affect the adsorption efficiency (Fig. 4a).

We thank the reviewer for his/her suggestion. The new data certainly helped improve our manuscript considerably.

(Comment 3) In Page 12, COP-99 do adsorb 12% of separation after 3h which is still significant. Can authors explain the adsorption? Does the COP-99 started expand overtime to accommodate the larger molecule? Performing longer experiments will provide some information. Additionally any other spectroscopic methods to elucidate this mechanism would be an ideal.

(Answer 3)

As the reviewer pointed out, COP-99 adsorbed 12 % of BPA over 3 h. The uptake of BPA is mainly attributed to its small size, well located within the range of accessible pore size distribution as

displayed in Figure 2. In fact, when compared to MB molecule, the theoretical maximum size of BPA is even smaller. Even then the adsorption of BPA on COP-99 is far slower than that of MB, in which MB exhibited a complete removal over 3 h while BPA showed just 12 % of removal for 3 h. We believe this is a direct evidence of fluorine effect of COP-99, facilitating the adsorption of charged molecules. Still, however, one might consider a number of factors that include (1) sieving, (2) hydrogen bonding between the hydroxyls of BPA and the fluorines on COP-99, and (3) adsorption on the grain surface cavities.

In order to further elucidate these points, we iterated that when the size of sorbate molecule exceeds the pore size threshold of COP-99, the adsorption does not happen in spite of the inherent charge of the molecules. We, therefore, conducted long-term adsorption test using RDB and BBG, which showed no uptake for 3 h of interaction, to confirm whether those molecules can occupy the pore voids over a long-term adsorption on COP-99 (Figure S10). After a 48 h of adsorption, RDB and BBG exhibited 27 % and 3 % of uptake on COP-99. Since the theoretical minimum size of RDB and BBG is slightly bigger than the pore threshold of COP-99, there is a chance of being taken in COP-99 for a long-time interaction. In particular, RDB with a size much closer to the average pore size of COP-99 could be adsorbed, showing a low but noticeable uptake of 27 %.

Figure S10. Long-term adsorption test of RDB and BBG using COP-99. Initial concentration (C_0) of all the dyes was adjusted to be 50 μM .

We thank the reviewer for his/her suggestion on mechanistic studies. First of all, in order to better understand the difference in IR spectra by the adsorption, we updated IR spectrum of Figure S3 (now Figure S3a) with the spectra of related monomers for a comparison. We note that FTIR

spectrum of COP-99 is now re-obtained on KBr disk (previously taken by ATR system) for better resolution. As displayed in the updated figure, COP-99 exhibited free O-H stretching vibration and H-bonded -OH vibration at 3670 cm^{-1} and 3300 cm^{-1} , respectively. The vibration at 1030 cm^{-1} is analysed by the comparison with FTIR spectra of two monomers. Hexafluorobenzene exhibited a strong C-F stretching at 1010 cm^{-1} region, while tetrafluorohydroquinone showed a vibration at 960 cm^{-1} coming from C-O stretching vibration. As the C-F and C-O vibrations are closely located even in monomers, and the strong peak at 1030 cm^{-1} on COP-99 is originated from the combination of C-F and C-O stretching vibrations. The weak vibration at around 740 cm^{-1} results from F-phenyl ring bending vibration (*Langmuir* 1998, 14, 1227; *Spectrochimica Acta* 1968, 24, 1999). Interestingly, after the adsorption of MB molecule on COP-99, the bending vibration at 740 cm^{-1} disappeared (Figure S3b). This implies that the specific binding of MB molecule on fluorine of COP-99 affects to the C-F bond angle, leading to the weakening of the C-F deformation vibration. We could not observe the change in 1030 cm^{-1} region after the MB adsorption, as the peak consisted of both C-F and C-O vibrations. We believe the change in FTIR spectrum explains in part the favourable interaction between fluorine and the MB molecule.

Figure S3. (a) FTIR spectra of COP-99 and related monomer units. (b) Change in FTIR spectrum before and after the adsorption of MB molecule.

We now include Figure S10 in the supplementary information, and the description regarding long-term adsorption of large-size molecules are added in the main text. The FTIR before and after MB adsorption is updated in the supplementary information, and the corresponding

discussion is included in the manuscript as follows:

(Page 14) On the other hand, RDB and BBG, which are larger than the average pore openings – despite the favourable fluorine-charge interaction-, did not show any uptake for 3 h of interaction. The latter observation further confirms that synergetic effect between narrow pore confinement and fluorine-cation interaction leads to a charge specific size selective adsorption behaviour. In order to further elucidate these points, we iterated that when the size of sorbate molecule exceeds the pore size threshold of COP-99, the adsorption does not go beyond surface coverage in spite of the inherent charge of the molecules. We, therefore, conducted long-term adsorption test using RDB and BBG, which showed no uptake for 3 h of interaction, to confirm whether those molecules can occupy the pore voids over a long-term adsorption on COP-99 (Supplementary Fig. 10). After a 48 h of adsorption, RDB and BBG exhibited 27 % and 3 % of uptake on COP-99. Since the theoretical minimum size of RDB and BBG is slightly bigger than the pore threshold of COP-99, there is a chance of being taken in COP-99 for a long-time interaction. In particular, RDB with a size much closer to the average pore size of COP-99 could be adsorbed, showing a low but noticeable uptake of 27 %. These size dependent observations may be attributed to the surface adsorption where RDB is packed in more finely than BBG.

(Page 11) The specific interaction between fluorine on COP-99 and MB molecule was confirmed by FTIR spectrum (Supplementary Fig. 3b). After the MB adsorption, COP-99 showed insignificant change in FTIR spectrum due to the weak electrostatic interaction between COP-99 and MB molecule. Nonetheless, there was one minor peak disappearing through the MB adsorption, which is at the position for F-phenyl ring bending mode at 740 cm^{-1} . This may be attributed to the hindrance of the C-F bond angle as the MB molecule favourably binds on the fluorine of COP-99.

We thank the reviewer again for taking time to contribute our study.

(Comment 4) When submitting for such high impact paper, the reviewer would like to see column breakthrough experiments with mixed organic molecules in the same solution instead of performing experiments individually. I highly recommend these experiments with variable concentrations. For example, trace amounts of 4 nitro phenol in low concentrations vs higher concentrations of larger organic molecules.

(Answer 4)

We thank the reviewer for this great suggestion. Based on the reviewer's recommendation, we conducted a column experiment using COP-99 to separate two tested molecules from their binary mixture. We chose 4-NP and BBG mixture as their maximum absorbance wavelength in UV spectrum is not overlapped to each other. A column for the separation test was made by a recipe reported elsewhere (*Nat. Commun.* 2014, 5, 5537). In short, about 150 mg of COP-99 was packed in a glass column with a diameter of 4 mm, sandwiched between cotton wool plugs. The total length of the column was 20 cm, and the packed sample length was about 3 cm. The dye mixture was prepared in a 1:1 v/v ratio of 50 μM BBG and 30 μM 4-NP-b, in which the smaller target molecule – 4-NP – was in lower concentration to simulate referred conditions. We couldn't do much higher concentration difference since the detection limits would create problems in the accuracy of the removal readings. As shown in Figure 7b, the UV spectra before and after column treatment exhibited that the dye mixture was clearly separated with a sharp decrease of 4-NP concentration. The concentration change was found to be 28 % and 99 % for BBG and 4-NP, respectively, indicating the separation of the two molecules. The BBG, despite of no uptake in a batch condition, showed a moderate uptake after the column, and it may be attributed to tight packing density of the column.

We now include a paragraph to describe a column experiment of COP-99 and added Figure 7b in the main text:

Figure 7. (b) UV-vis spectra of mixed dye solution before and after passing through a packed column of COP-99. The initial dye mixture was prepared in a 1:1 v/v ratio of 50 μM BBG and 30 μM 4-NP-b. *Inset* displays the photograph of the actual column tested in the study.

(Page 20) The COP-99 was further utilized in a column experiment to separate two molecules having different sizes. We chose 4-NP and BBG mixture as their maximum absorbance wavelength in UV spectrum does not overlap each other. A column for the separation test was made by packing about 150 mg of COP-99 in a glass column with a diameter of 4 mm, sandwiched with cotton wool. The total length of the column was 20 cm, and the packed sample length was about 3 cm. The dye mixture was prepared in a 1:1 v/v ratio of 50 μ M BBG and 30 μ M 4-NP-b, in which the smaller target molecule – 4-NP – was in lower concentration to simulate actual conditions. As shown in Fig. 7b, the UV spectra before and after column treatment exhibited that the dye mixture was clearly separated with a sharp decrease of 4-NP concentration. The concentration change was found to 28 % and 99 % for BBG and 4-NP, respectively, indicating the effective separation of the two molecules. The BBG, despite of no uptake in a batch condition, showed a moderate uptake after the column, and it may be attributed to tight packing density of the column. The low concentration uptake and column separation test have demonstrated the feasibility of COP-99 for the actual field applications.

(Page 23) For a column separation of dye mixture, about 150 mg of COP-99 was finely ground by a mortar and packed in a glass column, sandwiched between cotton wool plugs to prevent a mass loss. The inner diameter of the column was about 4 mm, and total length of the column was 20 cm. The packed sample length in the column was about 3 cm. Dye mixture for a column breakthrough was prepared in a 1:1 v/v of 50 μ M BBG and 30 μ M 4-NP-b. About 10 ml of dye mixture was passed through the column, and the effluent was collected and analysed by UV-Vis spectrophotometer to determine the concentration change.

We are very grateful to the reviewer for taking time to give insightful comments and contribute to our work by improving it greatly.

Reviewers' comments:

Reviewer #1 (Remarks to the Author):

Yavuz and coworkers have submitted a revised manuscript that describes a porous polymer capable that derives its unique selectivity based on its exposed organofluorine functionalities. I had a high opinion of the original submission, and I feel that the authors have done an extremely thorough and professional job for this revision. In my opinion, this article makes an important new contribution to fundamental knowledge of noncovalent interactions that are important for separations and leverages this knowledge to prepare an adsorbent material with interesting and promising performance. This advance should be reported in a top journal such as Nature Communications.

Reviewer #2 (Remarks to the Author):

In the revised manuscript, many additional experimental results have been added and the property of charge specific size dependent separation is more clearly illustrated relative to the original one. However, after careful evaluation of the revised manuscript and the rebuttal letter, as well as comparison with the published results, in my opinion, the breakthrough of this work still cannot meet the high standard of a high impact journal like Nature Communications. The reasons are listed as follows:

1. In the real-world application of the adsorbent for water treatment, it is interesting how low the target species concentration reaches. Along this line, the authors should give some studies on the performance of the material at extremely low concentration (e.g. ppb level).
2. The column breakthrough experiments have been added in the revised manuscript, but only two compounds with totally different properties have been tested, and only the beginning part of the solution after breakthrough has been tested. A more detailed breakthrough experiment should be given.
3. The solubility of the material in water should be tested in more detail, for example long term hydrothermal treatment are suggested, not just sonication for 1 min at 50 {degree sign}C. A picture is not sufficient, the filtrate after solubility test should be analyzed.

Reviewer #3 (Remarks to the Author):

After careful evaluation of the manuscript with additional experiments i would recommend the paper for publications. I trust my comments helped the authors to evaluate the hypothesis in terms of effect of hydrogen bonding, size/shape/charge separation.

Reviewers' comments:

Reviewer #1:

Yavuz and coworkers have submitted a revised manuscript that describes a porous polymer capable that derives its unique selectivity based on its exposed organofluorine functionalities. I had a high opinion of the original submission, and I feel that the authors have done an extremely thorough and professional job for this revision. In my opinion, this article makes an important new contribution to fundamental knowledge of noncovalent interactions that are important for separations and leverages this knowledge to prepare an adsorbent material with interesting and promising performance. This advance should be reported in a top journal such as Nature Communications.

We truly thank the reviewer for acknowledging the importance of our work, and recommending once again for the acceptance in Nature Communications.

Reviewer #2:

In the revised manuscript, many additional experimental results have been added and the property of charge specific size dependent separation is more clearly illustrated relative to the original one. However, after careful evaluation of the revised manuscript and the rebuttal letter, as well as comparison with the published results, in my opinion, the breakthrough of this work still cannot meet the high standard of a high impact journal like Nature Communications. The reasons are listed as follows:

We sincerely thank the reviewer for his/her comments and acknowledgment of the clearer discussion on the unusual charge specific size dependent separation. We think we addressed the queries that the reviewer has pointed out below. We hope that the reviewer is now satisfied with the additional experimental results and modifications.

1. In the real-world application of the adsorbent for water treatment, it is interesting how low the target species concentration reaches. Along this line, the authors should give some studies on the performance of the material at extremely low concentration (e.g. ppb level).

We thank the reviewer for the insightful comment. We agree that the monitoring of contaminant

concentration in ppb level is of importance for water treatment application, as water contaminants exist under ppm level in actual aqueous conditions. Therefore, we have now carried out batch adsorption tests of dye molecules in ppb level, and analysed the residual concentrations by HPLC. In addition to the NP-a sorption test (updated in **Figure 7a**), the small molecules showing a certain level of uptake using COP-99 were now tested, which are NP-b, MB and BPA. The initial concentration of the target molecules was adjusted to be about 100 ppb. The mobile phase for HPLC analysis varied depending on the target molecules, i.e., methanol 60%: water 40% for NP-b, acetonitrile 30 %: water 70 % acidified with H₃PO₄ to pH ~3.5 for MB, and acetonitrile 40 %:water 60 % for BPA. The intensity of the effluent UV absorbance was monitored at $\lambda = 335$ nm, 635 nm, and 276 nm for NP-b, MB, and BPA, respectively. LC calibration curve of the molecules was created for quantitative analysis using five standard concentrations of 0.05 ppm, 0.1 ppm, 10 ppm, 50 ppm, and 100 ppm.

Supplementary figure 16. LC spectra of (a) 4-NP-b, (b) MB and (c) BPA before and after the treatment with COP-99. RE stands for removal efficiency via the adsorption with COP-99.

The LC spectra of the tested dye substrates in ppb level exhibited a clear decline in their concentrations after the treatment of COP-99 (**Figure S16**). In the case of 4-NP-b, the initial solution showed a peak at 2.17 min of retention time, and after being treated with COP-99 for 1 h, the peak intensity decreased down at 2.17 min of retention time (**Figure S16a**). The concentration of 4-NP-b, analysed with LC calibration curve (**Figure S14b**), changed from 140.35 ppb to 18.27 ppb, indicating 87 % of removal efficiency. The adsorption efficiency of 4-NP-b was slightly lower than that of 4-NP-a, and this is in accordance with the batch adsorption results in higher concentrations (**Figure 4a**) where the anionic nature of 4-NP-b slightly obstructs the adsorption on the fluorinated surface of COP-99. As displayed in **Figure S16b**, MB exhibited a complete removal in ppb level. Before adsorption, it showed a sharp peak at 1.83 min of retention time and the peak is beyond the detection limit after the adsorption with COP-99 for 3 h. On the other hand, BPA, evolved at 3.95 min of retention time, exhibited the concentration change from 134.19 ppb to 55.58 ppb, revealing 58.58 % of removal efficiency (**Figure S16c**). The slower uptake of BPA confirms the charge-dependent adsorption behaviour of COP-99. Therefore, COP-99 clearly demonstrates expected adsorption performance for small and charged molecules even at low concentrations.

We now include **Figure S16** in the supplementary information, and added the description in the manuscript as follows. The HPLC calibration curves are also updated in **Figure S14** accordingly.

(Page 20) Other small molecules prepared in ppb levels, i.e. 4-NP-b, MB and BPA, also exhibited the similar patterns of adsorption as observed for adsorption in the high concentrations (**Supplementary Fig. 16**). Particularly, MB showed no residual peak on LC spectrum, indicating a complete removal after treatment with COP-99 for 3 h (**Supplementary Fig. 16b**).

(Page 24) The mobile phase for HPLC analysis varied depending on the target molecules, i.e., methanol 60%: water 40% for NP-b, acetonitrile 30 %: water 70 % acidified with H₃PO₄ to pH ~3.5 for MB, and acetonitrile 40 %: water 60 % for BPA. The intensity of the effluent UV absorbance was monitored at $\lambda = 335$ nm, 635 nm, and 276 nm for NP-b, MB, and BPA, respectively.

Supplementary figure 14. HPLC calibration curves of the tested dye substrates from five concentrations of 0.05 ppm, 0.1 ppm, 10 ppm, 50 ppm, and 100 ppm. (a) 4-NP-a, (b) 4-NP-b, (c) MB, and (d) BPA.

We thank the reviewer for his/her suggestion to improve our study.

2. The column breakthrough experiments have been added in the revised manuscript, but only two compounds with totally different properties have been tested, and only the beginning part of the solution after breakthrough has been tested. A more detailed breakthrough experiment should be given.

We thank the reviewer for bringing this point to our attention. With all due respect, our aim in this study is to show our discovery of charge specific size dependent separation. Per the reviewer's request, however, we carried out column separation experiments, with the capabilities at hand. But as the reviewer would agree, there are more factors in a column setup than just chemical interactions. A

typical breakthrough test would need much detailed controls in engineering parameters such as granule size, column bed volume, bed height, packing density, and flow rate [*Ecol. Eng.*, **2014**, 73, 270-275]. We are now proactively looking for the best breakthrough test set-up, and hope that the new data will prepare this discovery to the wide scope of application in real life. This is why we are now using the phrase ‘column separation’ throughout the manuscript to refer to our new experiments.

Per the reviewer’s request, we would like to clarify our findings in a set of more column separation tests. We previously chose 4-NP and BBG as a dye mixture since they have the biggest size difference among all the dye substrates (**Figure 2**) and their maximum absorbance of UV-Vis spectrum does not interfere each other, so that we could easily observe size-dependent separation behaviour through COP-99 column. Furthermore, the contact time through a column is much shorter than that of batch adsorption test, and we had to select a pair of which one is adsorbed very quickly (i.e., 4-NP) and the other is slowly or rarely adsorbed (i.e., BBG). The effluent concentration after the column separation was analysed after filtering 10 mL of dye mixture (**Figure 7b**), and as the reviewer pointed out, 10 mL would be the very beginning part of the effluent to check the breakthrough point. When the column was fed up to 60 mL accumulatively in the same column, the percentage of removal of 4-NP and BBG stayed as 98.8 % and 28.4 %, respectively, and up to 100 mL, the removal efficiency became 95.4 % and 16.5% for 4-NP and BBG, respectively. This extended treatment confirms the previous observation that there is a clear trend of size-dependent separation via COP-99 column. We note that, however, the breakthrough test should be conducted in more controlled manner in a future study.

We observed that the larger BBG dye was slightly removed after being filtered through the COP-99 column, and we believe this is due to compact packing density of the column. Thus we carried out batch adsorption test of the dye mixture to confirm the selective adsorption behaviour of COP-99. The batch adsorption was done by soaking 8 mg of COP-99 in 8 mL of dye mixture for desired time period. As displayed in the figure below (**Figure S17c**), the smaller 4-NP was completely removed from the mixture over 6 h of soaking while larger BBG exhibited no concentration change during the treatment. This result verifies that the moderate uptake of BBG through the column separation is primarily attributed to the column packing condition. Such a difference in adsorption

behaviour between a column and a batch system was also observed in the literature [*Angew. Chem. Int. Ed.*, **2015**, 54, 1-6].

The adsorption test with 4-NP/BBG mixture led us to understand size-dependent adsorption behaviour of COP-99, and in order to identify charge-based adsorption capability of COP-99, we prepared a new test pair: mixed solution of MB and BPA. The dye mixture was made in a 1:1 v/v ratio of 50 μM MB and 50 μM BPA. Firstly, the binary solution was filtered through COP-99 column prepared in the same way described earlier, and the effluent was collected up to 40 mL and the concentration change was analysed by using UV-Vis spectrophotometer. When the MB/BPA mixture was filtered through the COP-99 column, both molecules exhibited concentration decrease in a similar level, showing the removal efficiency of 47 % and 55.4 % for MB and BPA, respectively (**Figure S17b**). We assume that this is originated from the column packing condition as we have seen the separation test result from 4-NP/BBG mixture. The size-dependent separation was dominant than the charge-based separation when the mixed molecules pass through a densely packed column, and MB and BPA having almost similar molecular size were filtered out at the same time. While there was no selective adsorption in the column separation, COP-99 showed charge-selective uptake of MB out of the MB/BPA dye mixture in batch adsorption test. After immersing 8 mg of COP-99 in the 8 mL of MB/BPA mixed solution for 12 h, MB was removed up to 87 % and BPA was adsorbed 22 %, indicating the fluorine-charge interaction facilitates selective adsorption of MB out of the mixed solution (**Figure S17d**). Thus, we believe that batch adsorption tests using dye mixtures verify our main findings of size- and charge-selective adsorption behaviour of fluorinated COP-99.

We now include **figure S17** in the supplementary information and describe the batch adsorption (soaking) test in the main text as follows:

Supplementary figure 17. UV-Vis spectra of mixed dye solutions of (a,c) 4-NP/BBG and (b,d) MB/BPA (a, b) before and after being treated via a packed column of COP-99 and (c, d) during soaking COP-99 in the solutions.

(Page 20) ...and it may be attributed to tight packing density of the column. In fact, batch adsorption test with the 4-NP/BBG mixture exhibited that the smaller 4-NP was completely removed from the mixture over 6 h of soaking while larger BBG did not show change in concentration during the treatment (**Supplementary Fig. 17c**).

(Page S24) *Note:* Along with 4-NP/BBG mixture shown in **Fig. 7b**, mixture of MB and BPA having similar molecular size was also tested for column separation and batch adsorption. When MB/BPA mixture was filtered through the COP-99 column, both molecules exhibited concentration decrease in a similar level, showing the removal efficiency of 47 % and 55.4 % for MB and BPA, respectively (**Figure S17b**). We believe this is originated from the column packing condition as we have seen the

separation test result from 4-NP/BBG mixture. The size-dependent separation was dominant than the charge-based separation when the mixed molecules pass through a densely packed column, and MB and BPA having almost similar molecular size were filtered out at the same time. While there was no selective adsorption in the column separation, COP-99 showed charge-selective uptake of MB out of the MB/BPA dye mixture via batch adsorption. After immersing 8 mg of COP-99 in the 8 mL of MB/BPA mixed solution for 12 h, MB was removed up to 87 % and BPA was adsorbed 22 %, indicating the fluorine-charge interaction facilitates selective adsorption of MB from the mixed solution (**Figure S17d**).

We, once again, thank the reviewer for improving our manuscript.

3. The solubility of the material in water should be tested in more detail, for example long term hydrothermal treatment are suggested, not just sonication for 1 min at 50 oC. A picture is not sufficient, the filtrate after solubility test should be analyzed.

We thank the reviewer for the constructive criticism. Per the reviewer's suggestion, the solubility of COP-99 was tested under hydrothermal conditions. In order to prevent from the possible evaporation of D₂O under the elevated temperature, we carried out the test in closed glass ampoule. A glass ampoule also provides high pressure; conveniently representing the harshest conditions a water treatment sorbent can be subjected to. In a typical experiment, ~20 mg of COP-99 was placed in a Pyrex ampoule (5 mL) and 1.5 mL of D₂O was transferred in the ampoule. The reaction mixture was frozen in liquid nitrogen, further evacuated and flame-sealed. After being warmed, the ampoule was treated at 110°C for 24h, and the reaction filtrate was analysed by NMR spectroscopy. As displayed in the figure below, we observed that COP-99 did not exhibit any colour change and the D₂O filtrate also stayed transparent after the hydrothermal treatment (**Figure S9d**). The ¹H NMR spectrum of the D₂O filtrate only showed a D₂O residual peak, and the ¹³C NMR spectrum also displayed no peaks, indicating the COP-99 is insoluble in water even at a boiling condition (**Figure S9e** and **S9f**, respectively). In addition, a structural disintegration must have yielded carbon-containing species, and we are happy for not seeing any carbon fragments in the highly sensitive ¹³C NMR.

In the ^{19}F NMR spectrum (**Figure S9g**), a small peak at -130.27 ppm, corresponding to aqueous F^- ions (Aqueous F^- of $\text{KF} = -125.3$ ppm). This implies that fluorines on COP-99 were detached from the polymer network at the elevated temperature and pressure, and we assumed that D_2O replaces few fluorines to produce deuterium fluoride at high temperature via nucleophilic substitution. Therefore, we further conducted the aqueous treatments at lower temperature, i.e., 50°C . As expected, there were no peaks in ^1H , ^{13}C , ^{19}F NMR spectrum from the D_2O filtrate treated at 50°C for 60 h as nucleophilic substitution would not happen at low temperature (**Figure S9h**). From quantitative elemental analysis, we found that the fluorine content has been decreased 4.6 % after the boiling test at 110°C (only 20 % of all fluorines) due to the fluorine exchange with D_2O ; however, there was no noticeable change in fluorine content after the treatment at 50°C even after 60 hours. This indicates that the nucleophilic substitution of fluorines is possible (albeit minor) under boiling condition, but the fluorine content was not changed at 50°C as the nucleophilic substitution reaction does not happen at this temperature. In summary, the COP-99 was stable in water up to at least 50°C with no change in the amount of fluorines and its framework is stable even at high temperature and pressures. Temperature of wastewater is known to never above 50°C [*Clean Techn. Environ. Policy*, **2005**, 7, 198-202], thus COP-99 should be safe to be utilized in water treatment application owing to its durability and insolubility in water.

Figure. ^{19}F NMR spectrum of D_2O filtrate after being treated at 50°C for 60 h.

We now updated **Figure S9** and added the description in the manuscript as follows:

Supplementary figure 9. (a) Solubility test of COP-99 in common solvents. Images were taken after a sonication for 1 min at 50 °C. (b) Contact angle for a water droplet on the surface of COP-99. (c) Photograph of COP-99 in water. Owing to the hydrophobicity, COP-99 floats on the surface of water. (d) Long-term boiling test of COP-99 (20 mg) in D₂O (1.5 mL) at 110°C for 24 h. In order to prevent D₂O evaporation, the test was conducted in a sealed glass ampoule. There was no noticeable change in both COP-99 and D₂O solution after boiling. (e) ¹H and (f) ¹³C NMR spectrum of the D₂O filtrate. (g) ¹⁹F NMR spectrum of filtrate exhibited the existence of free fluorine partially detached from COP-99 via nucleophilic substitution. (h) Fluorine peak was not observed when the COP-99 was treated in D₂O at 50 °C for 60 h.

(Page 6) Due to the highly cross-linked network structure, COP-99 was insoluble in common solvents (Supplementary Fig. 9a) and boiling water (Supplementary Fig. 9d, e, and f).

(Page S15) *Note:* Long-term boiling test of COP-99 was carried out under hydrothermal condition. In order to prevent from the possible evaporation of D₂O under the elevated temperature, we carried out the test in closed glass ampoule. Typically, about 20 mg of COP-99 was placed in a Pyrex ampoule (5 mL capacity) and 1.5 mL of D₂O was transferred in the ampoule. The reaction mixture was frozen in liquid nitrogen, further evacuated and flame-sealed. After being warmed, the ampoule was treated at

110°C for 24h, and the reaction filtrate was analysed using NMR spectroscopy.

We observed that COP-99 did not exhibit any colour change and the D₂O filtrate also stayed transparent after the hydrothermal treatment (**Supplementary figure 9d**). The ¹H NMR spectrum of the D₂O filtrate only showed a D₂O residual peak, and the ¹³C NMR spectrum also displayed no peaks, indicating the COP-99 is insoluble in water even at a boiling condition (**Supplementary figure 9e** and **9f**, respectively). In the ¹⁹F NMR spectrum (**Supplementary figure 9g**), however, showed a small peak at about -130.27 ppm, corresponding to aqueous F⁻ ions (e.g. Aqueous F⁻ of KF = -125.3 ppm). This implies that few fluorines on COP-99 were detached from the polymer network at the elevated temperature, and we assumed that D₂O replaces fluorines to produce deuterium fluoride in high temperature via nucleophilic substitution. When COP-99 was treated under lower temperature, i.e., 50 °C for 60 h, there were no peaks in ¹H, ¹³C, ¹⁹F NMR spectrum in the D₂O filtrate (**Supplementary figure 9h**). We found that the fluorine content has been decreased 4.6 % after the boiling test at 110 °C (only 20 % of all fluorines) due to the fluorine exchange with D₂O, however, there was no noticeable change in fluorine content after the treatment at 50 °C. This indicates that the nucleophilic substitution of fluorines is plausible (albeit minor) under boiling condition, but the COP-99 was stable in water at least up to 50 °C with no change in the amount of fluorines and its framework is stable even at high temperature and pressures. Therefore, COP-99 should be safe to be utilized in water treatment application owing to its stability and insolubility in water.

We appreciate the reviewer for his/her suggestions to clarify our results. We believe these comments and others have helped our study to become complete.

Reviewer #3:

After careful evaluation of the manuscript with additional experiments I would recommend the paper for publications. I trust my comments helped the authors to evaluate the hypothesis in terms of effect of hydrogen bonding, size/shape/charge separation.

We sincerely thank the reviewer for his/her insightful comments that significantly improved our study.

And we are indebted for his kind recommendation for publication.

REVIEWERS' COMMENTS:

Reviewer #2 (Remarks to the Author):

The authors conducted additional experiments to address the reviewer's comments, and the manuscript is much improved in comparison with the previous versions. This reviewer lends the decision to the editor whether or not this work is suitable for publication in Nature Communications.

Reviewer #2 (Remarks to the Author):

The authors conducted additional experiments to address the reviewer's comments, and the manuscript is much improved in comparison with the previous versions. This reviewer lends the decision to the editor whether or not this work is suitable for publication in Nature Communications.

RESPONSE: We are indebted to the reviewer for his/her comments to improve our study.